# Kronos scRT: a uniform framework for single-cell replication timing analysis

Stefano Gnan [1], Joseph M. Josephides [1], Xia Wu[1,4], Manuela Spagnuolo[1], Dalila Saulebekova[1], Mylène Bohec[2], Marie Dumont[3], Laura G. Baudrin[2], Daniele Fachinetti [3], Sylvain Baulande [2] & Chun-Long Chen [1✉]

Mammalian genomes are replicated in a cell type-specific order and in coordination with transcription and chromatin organization. Currently, single-cell replication studies require individual processing of sorted cells, yielding a limited number (<100) of cells. Here, we develop Kronos scRT, a software for single-cell Replication Timing (scRT) analysis. Kronos scRT does not require a specific platform or cell sorting, which allows investigating large datasets obtained from asynchronous cells. By applying our tool to published data as well as droplet-based single-cell whole-genome sequencing data generated in this study, we exploit scRT from thousands of cells for different mouse and human cell lines. Our results demonstrate that although genomic regions are frequently replicated around their population average RT, replication can occur stochastically throughout S phase. Altogether, Kronos scRT allows fast and comprehensive investigations of the RT programme at the single-cell resolution for both homogeneous and heterogeneous cell populations.

---

[1] Institut Curie, PSL Research University, CNRS UMR3244, Dynamics of Genetic Information, Sorbonne Université, 75005 Paris, France. [2] Institut Curie, Genomics of Excellence (ICGex) Platform, PSL Research University, 75005 Paris, France. [3] Institut Curie, PSL Research University, CNRS UMR144, Cell Biology and Cancer, 75005 Paris, France. [4] Present address: Zhongshan School of Medicine, Sun Yat-sen University, Guangzhou, Guangdong 510080, China. ✉email: chunlong.chen@curie.fr

DNA replication is a fundamental process in all living organisms that guarantees genome duplication before cell division. Based on inter-origin distance (one origin every ~100 kb in mammalian cells) and replication fork speed (1–3 kb/ min), if all replication origins were simultaneously activated in a mammalian cell, it would only take about 30 min to complete the genome replication[1,2]. However, due to limiting factors of replication initiation and replication fork progression, DNA replication is not simultaneously initiated at all potential origins[3,4]. Rather, each cell type displays a defined selection and temporal order of origin firing, and DNA replication of a mammalian genome is completed in several hours (usually between 6 and 12 h)[5,6]. Furthermore, this temporal and spatial organization is coordinated with other processes, such as chromatin organization and gene transcription[7–9]. The cell type-specific programme that regulates DNA replication during the synthesis phase (S phase) is referred to as replication timing (RT) programme[10,11]. This programme may be altered in human diseases, such as cancer and neurological disorders[12–14]. In addition, we and others have shown that RT plays an important role in shaping the mutational landscape and in impacting genome stability in both normal and cancer cells[15–19]. Therefore, RT is an important feature to better understand the underlying causes or the outcomes of genomic instability.

In the last decade, high-throughput single-cell omics have allowed the study of intercellular variability and shed light on the cell functional and structural dynamics. Advances in high-throughput single-cell sequencing techniques offered the possibility to analyze RT at the single-cell level. Compared with bulk cell studies, recent single-cell RT (scRT) studies[20–23] investigated the RT programme in individual cells and RT variability among cells. However, all published scRT studies required the identification of the cell phase (i.e. G1, S) by fluorescence-activated cell sorting (FACS) and manual processing of individual cells to perform single-cell Whole-Genome Sequencing (scWGS). These steps limit the sample size to only tens to a hundred cells[21–23], which leads to scalability concerns. Nevertheless, the investigations of the obtained scRT data in these studies support the hypothesis that most replication domains follow the pre-determined RT programme in the same population[21–23]. However, there is still room to explore non-conforming events in single cells that deviate from the population average RT and may undergo stochastic replication. A recent important advancement in Optical Replication Mapping (ORM) allows mapping newly replicated DNA and thus, tracking early initiation events at the single-molecule level[24]. ORM analysis of individual initiation events in HeLa cells synchronized at the very beginning of S phase showed that although most early initiation events occur in the early-replicating regions of the genome, a significant number (~9%) happen in the late-replicating regions. This finding supports a stochastic model of replication initiation. Unfortunately, these rare stochastic RT events cannot be correctly identified in the currently available scRT data due to their limited sample sizes.

Here, to overcome these limitations, we present a uniform computational framework named Kronos scRT to investigate scRT based on single-cell copy number variation (scCNV) detection in scWGS data. Our pipeline can be used to analyze datasets obtained from FACS-sorted cells or directly from asynchronously growing cells by single-cell whole-genome amplification (scWGA), the droplet-based 10x Genomics Chromium scCNV Solution, single-cell High-throughput Chromosome conformation capture (scHi-C), among other related data. This pipeline allows a ten-fold increase in the number of cells used to analyze scRT (>1000 cells in one experiment) compared with previous scRT studies. By analyzing published data and the droplet-based scWGS data generated in the current study, we obtained large amounts of scRT data for different mammalian cell lines (up to 1353 S-phase cells for a given cell type from a single experiment; 4724 cells analyzed in total). These data allowed us to construct the S-phase progression trajectories of different cell types and to identify coexisting sub-populations. In addition, the analysis of significantly more cells enabled us to study DNA replication heterogeneity with unprecedented detail. We found that the observed scRT distribution is consistent with stochastic models of replication control. Replication kinetic modelling showed that measuring the firing efficiency in early S phase can predict the average firing time in a cell population. Here, we extended a previous single-molecule analysis to the single-cell level and showed that stochastic regulation of replication kinetics is a key feature of eukaryotic replication.

## Results

**Kronos scRT: a computational tool for scRT studies.** We developed Kronos scRT (https://github.com/CL-CHEN-Lab/Kronos_scRT), a tool that computes scRT under a unified framework and in a comprehensive manner (Fig. 1a and Supplementary Fig. 1a). First, single-cell DNA sequencing reads are aligned to the reference genome and counted in regular bins (20 kb in our analysis). This bin size can be adjusted in function of the mean coverage of the experiment. Read counts are then corrected for GC content and mappability bias. An option to blacklist genomic regions is also available (see "Methods"). Data are then segmented and copy number variation (CNV) is estimated for each individual cell. At this stage, two additional parameters are calculated: (i) the cell ploidy as the cell-weighted mean copy number (CN), and (ii) the intracellular bin-to-bin variability as the Depth Independent Median Absolute deviation of Pairwise Differences (DIMAPD) ("Methods"). In the final step, depending on the data type and the available information, Kronos scRT proposes different approaches to distinguish between cells in G1/G2 and in S phase. If cells are FACS-sorted in discrete populations, as in previous scRT studies[21–23], the cell phase information can be directly used to label cells into these two groups. For unsorted cycling populations, S-phase cells can be automatically detected based on two assumptions: (i) most cells belong to the G1/G2 population; (ii) the intracellular bin-to-bin variability is minimal in G1 and G2 cells (where all bins have similar CNs) and maximal in mid S-phase cells due to the asynchronous replication of adjacent bins (Fig. 1b, c). The program fits the variability data into a Gaussian distribution and identifies S-phase cells as outliers ("Methods"). Moreover, if the cell cycle distribution has been altered (i.e. S-phase enrichment without complete removal of G1- and G2-phase cells), users can manually set a variability threshold based on the data visual inspection (Fig. 1b).

Due to the method used for CN calculation ("Methods"), it is impossible to discriminate between G1 and G2 cells. This also has an effect on the S-phase population that is split into two groups. Cells in the first S-phase group have higher ploidy than cells in the G1/G2 phase and their bin-to-bin variability positively correlates with ploidy. Cells in the second S-phase group display lower ploidy than the G1/G2 pool and their bin-to-bin variability decreases as they approach the G1/G2 population (Fig. 1b). Therefore, before proceeding with the downstream analysis, S-phase cell ploidy must be adjusted either automatically (Fig. 1c) or manually with parameters imposed by the user ("Methods").

The data shown in Fig. 1b–d were obtained by analysis of a dataset of MCF7 breast cancer cells generated using the 10x Genomics microfluidic system. According to the American Type Culture Collection, this cell line has 80 autosomes in G1 (ploidy: 3.64), in agreement with the Kronos scRT estimation (Fig. 1b, c).

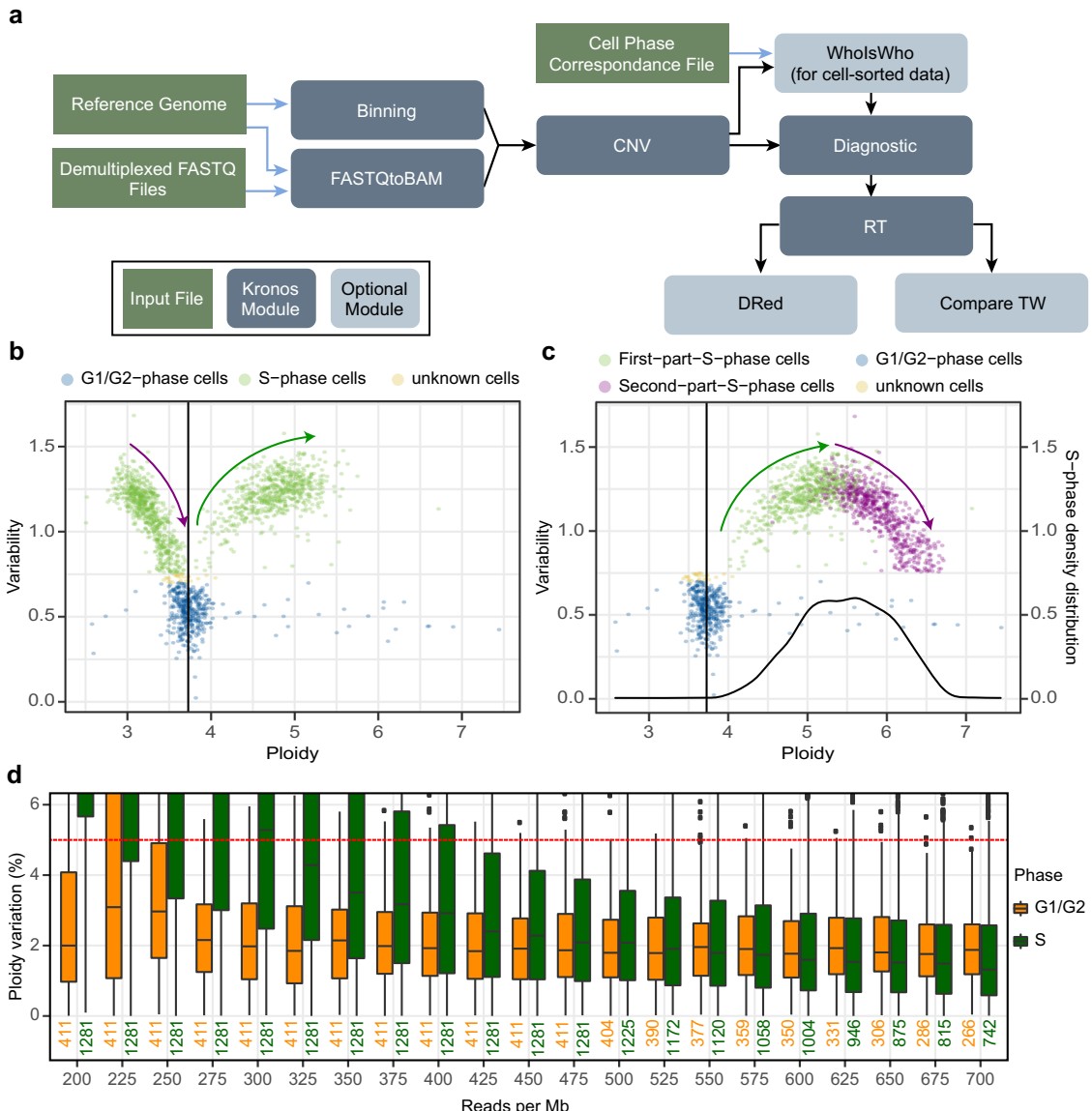

**Fig. 1 An efficient uniform framework for scRT extraction. a** The pipeline of Kronos scRT with its different modules. The input files, the main modules of Kronos RT, and the optional modules are shown in green, dark blue-grey, and light blue-grey, respectively. **b** Scatter plot reporting the mean cell ploidy on the x-axis and the bin-to-bin intracellular variability on the y-axis. Each point is a single cell and the colour is assigned based on a cut-off automatically calculated, or manually imposed, relative to the population variability ("Methods"). High variability is associated with S-phase cells (green), while low variability is associated with G1/G2-phase cells (blue). An unknown region can be manually set (yellow). The vertical black line represents the median ploidy of the population. The data are from asynchronous S-phase enriched MCF7 cells. The green and purple arrows show the S-phase progression of the first and second S-phase cell group, respectively ("Methods"). **c** Data presented in (**b**) after S-phase progression correction. The colour of the second S-phase group has been changed to purple. As in (**b**), the green and purple arrows indicate the S-phase progression. The black curve reports the S-phase density distribution that is used to calculate the parameters to adjust the S-phase progression ("Methods"). **d** Read down-sampling for the MCF7 cell dataset obtained using the 10x Genomics System. G1/G2- and S-phase cells are plotted in orange and green, respectively. Boxplots show the mean ploidy variation (in percentage) after down-sampling compared with the original value. The red dashed line indicates a change of 5%. Cells with more than 450 reads per Mb were used for the down-sampling. The numbers of cells (*n*) used for each down-sampling are reported below each boxplot, cells are analyzed over one experiment. In the boxplots, bounds of box: 25th and 75th percentiles; centre line: median; lower (upper) whisker: lower (upper) bound - (+) 1.5 x Interquartile range (IQR).

Following the ploidy estimation step, cells with low coverage are filtered out. The minimum number of reads required for CNV calling depends on the experimental settings and the procedure used to create the scWGS libraries. Therefore, for each dataset, it is important to select G1/G2- and S-phase cells with relatively high coverage to determine by down-sampling the robustness of CNV detection and the minimum number of reads required for correct CNV calling. Based on this down-sampling analysis, we

discovered that the ploidy estimation of G1/G2-phase cells was less sensitive to coverage changes than that of S-phase cells (Fig. 1d and Supplementary Fig. 1b, c). Then, from the down-sampling values, we selected a coverage threshold that did not allow the ploidy of 75% of S-phase cells to deviate more than 5% from the original ploidy estimation (Fig. 1d and Supplementary Fig. 1b, c). We estimated that in our MCF7 cell dataset, a coverage threshold of 117 reads per megabase (RPMb) per haploid genome

(425 RPMb/3.64 ploidy) was needed for a robust mean ploidy estimation.

Then, the adjusted CNs of cells that passed the filter can be used to calculate the scRT profiles. Based on the coverage of our data, the genome was binned into 200 kb non-overlapping windows and the weighted median CN for each cell was calculated. Using the G1/G2-phase cell population, a median CN profile was calculated and used to normalize the CN of each S-phase cell as a log2 ratio ("Methods"). Data were then binarized to obtain the scRT profiles, where 1 corresponds to replicated regions and 0 to non-replicated regions. This is based on the assumption that the CN is doubled in replicated regions of S-phase cells compared with the G1/G2 fraction, and therefore the log2 ratio is close to 1. Conversely, non-replicated regions will have the same CN as the G1/G2 population and a log2 ratio close to 0. Binarization is independent for each cell and is based on the identification of a normalized CN threshold. This is the threshold that minimizes the Euclidian distance between the generated scRT profiles and the original data ("Methods"). As an additional quality control, a pairwise Simple Matching Coefficient of the scRT profiles is calculated and cells with an RT profile that deviates from the main population are filtered out ("Methods"). Finally, the pseudo-bulk RT can be computed as the weighted mean of all scRT profiles and can be compared with the bulk RT ("Methods").

In conclusion, by combining CNV calling, inter- and intra-cellular variability, and quality control filtering, we successfully developed, Kronos scRT, a unified and efficient genome-wide scRT computational profiling tool for scRT studies.

**scRT analysis from scWGS data of sorted S-phase cells.** scRT studies usually require cell sorting by FACS due to the absence of in silico cell phase separation tools. In our pipeline, this cell phase information can be integrated in the analysis to label cells using the WhoIsWho module (Fig. 1a). To test the applicability of the Kronos scRT framework, we used previously published scWGS data derived from the sorted G1 and sorted mid S-phase cells of mouse embryonic stem cells (mESC, $n = 67$ cells) and of neuroectoderm cells obtained by differentiating mESCs to epiblast-like cells for 2 days followed by five days of embryoid body culture (hereafter called NE-7d cells, $n = 45$ cells)[22]. We determined the scRT profiles of these two cell types using Kronos scRT, and their cell phases were assigned using the FACS metadata (Supplementary Tables 1 and 2).

To demonstrate that Kronos scRT can efficiently detect scRT profiles, even with a small number of sorted mid S-phase cells, for both cell lines, we calculated the correlation between the pseudo-bulk RT, determined with our tool, and the bulk RT data generated by Takahashi and colleagues by immunoprecipitation of BrdU labelled newly replicated DNA (BrdU-IP) from early and late S-phase cells[22] (Fig. 2a). The high Spearman correlation values (0.890 for mESCs and 0.902 for NE-7d cells) (Fig. 2b and Supplementary Fig. 2a) demonstrated the robustness of our method and computational pipeline. In addition, in agreement with previous studies[21–23], the obtained mid-S binary replication signals showed that cell-to-cell variability exists, but is limited, and that RT organization is largely conserved in single cells.

We then wanted to quantify the replication variability within each cell population. As scRT data do not provide precise information on the actual time needed for genome replication, we decided to quantify scRT variability using the concept of $T_{width}$ introduced in a recent scRT study[21]. $T_{width}$ is defined as the time needed for a given genomic region to be replicated from 25% to 75% of cells in an S phase lasting 10 h. Although S-phase length is not the same in all cells, assuming a uniform 10h S-phase length

makes it easy to compare results of different datasets and results obtained in previous studies. We found that the $T_{width}$ ranged between 2.78 h and 2.81 h in mESCs and between 2.76 h and 2.37 h in mNE-7d cells, for early and late-replicating regions, respectively. Using the Compare TW module, we performed a bootstrap-based null hypothesis test with $H_0$: $T_{width\_early} = T_{width\_late}$ and with $H_1$: $T_{width\_early} \neq T_{width\_late}$ ("Methods", Fig. 2c). We did not observe any significant statistical difference between early and late $T_{width}$ values ($p = 0.43$) in mESCs. Conversely, in mNE-7d cells, late-replicating regions were less variable than early ones ($p < 10^{-4}$). These results suggest that in mESCs, as observed by Dileep & Gilbert[21], RT does not present significant differences between early and late S-phase, while in mNE-7d cells, RT shows lower variability at the end of the S phase.

**Determination of scRT data from asynchronous cells using a microfluidic-based system.** To demonstrate that Kronos scRT can detect scRT without cell sorting and cell phase identification (e.g. by FACS), we first generated scWGS data for 368 oestrogen-treated cycling MCF7 cells (containing about 20% of S-phase cells) using the droplet-based 10x Genomics Chromium scCNV Solution (Fig. 3a, Supplementary Table 1). As previously discussed, this allowed us to automatically identify S-phase cells (Fig. 3a, left panel). Moreover, the even cell distribution across the S phase allowed using the automatic S-phase correction (Fig. 3a, right panel). We calculated the scRT and pseudo-bulk RT for 82 identified S-phase cells (Fig. 3b and Supplementary Fig. 2b). The pseudo-bulk RT was highly correlated (Spearman correlation $R = 0.913$) with the bulk RT (Fig. 3c). By visual inspection of the scRT profiles, we could clearly identify some variability (Fig. 3b). The even distribution of cells throughout the S phase allowed us to perform the $T_{width}$ analysis with higher resolution than that of sorted mid S-phase cells (Fig. 3d). As observed in previous studies[22,25], $T_{width}$ values were smaller at the beginning and at the end of the S phase, and progressively increased towards the mid S phase (Fig. 3d). Although only 82 cells were used in the analysis, such results support the hypothesis that the initiation and termination of the replication programme are more tightly regulated than mid S-phase replication.

**scCNV/scRT analyses allow identifying cell sub-populations within a heterogeneous population.** To obtain a more representative estimation of cell-to-cell RT variability, we performed scWGS analysis of a cell population that contained more S-phase cells. As unsorted cell populations generally contain more G1/G2-than S-phase cells, to reduce the sequencing cost, we increased the number of S-phase cells in the MCF7 samples to be sequenced by FACS sorting using a gate that contains the majority of the S-phase cells ("Methods"). Using this approach, we obtained 1777 MCF7 cells, most of which ($n = 1353$) were in S-phase (Fig. 1b, c, Supplementary Table 1).

While performing the dimensionality reduction analysis with Uniform Manifold Approximation and Projection (UMAP)[26,27] on the scRT profiles of the S-phase enriched MCF7 cells, we noticed that cells were distributed into two distinct trajectories in which cells were sorted according to their S-phase progression (Supplementary Fig. 3a). We hypothesized that the presence of two RT groups was probably due to the existence of two cell sub-populations with different chromosomal rearrangements. Therefore, we further performed the dimensionality reduction analysis using the scCNV data, instead of the scRT data, for G1/G2- and S-phase cells. We observed a clear separation into four groups (two G1/G2-phase groups and two S-phase groups) and in two sub-populations (Fig. 4a). The mean G1/G2 ploidy was ~3.73 for

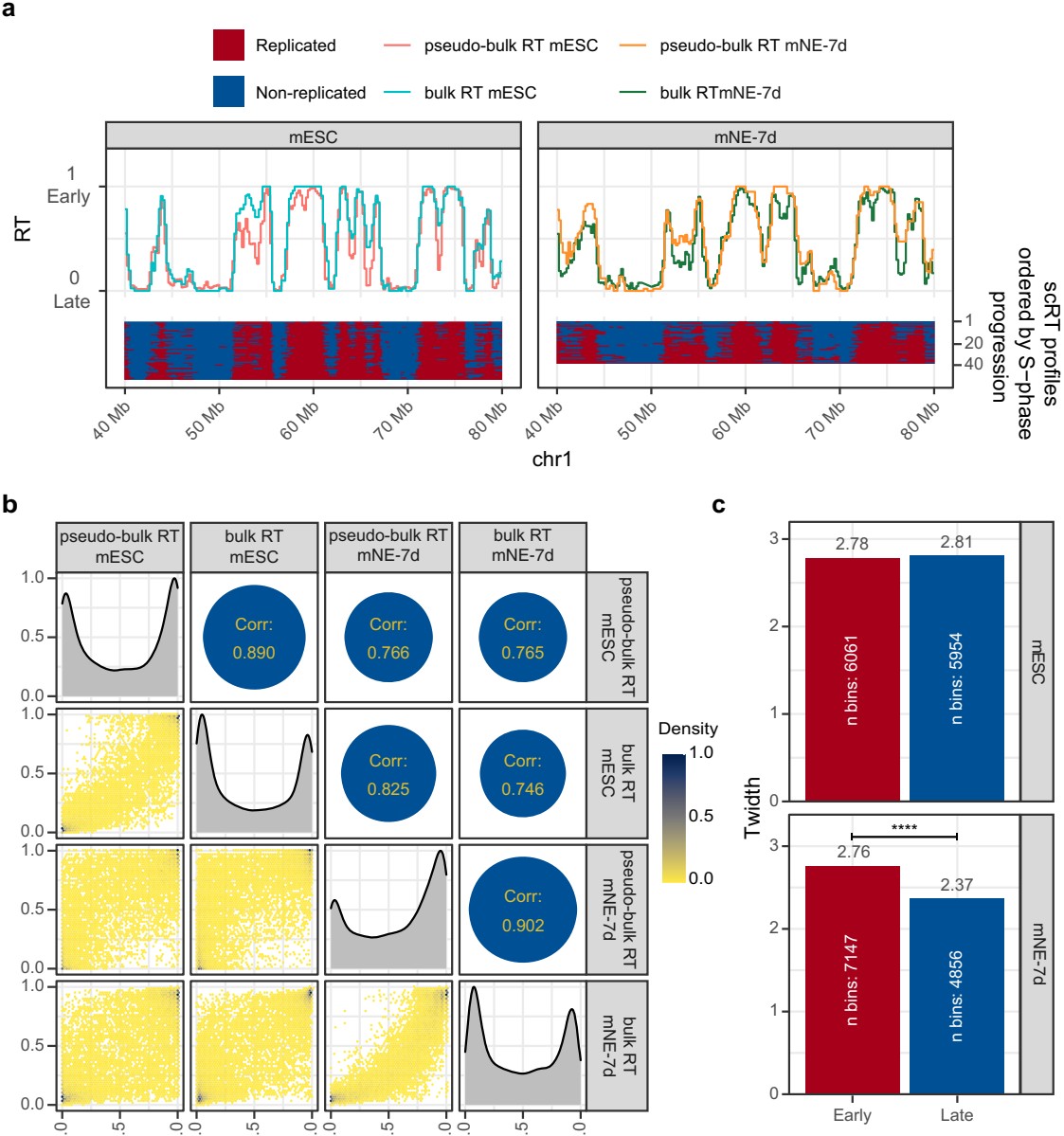

**Fig. 2 Extracting scRT from sorted mid S-phase mouse cells. a** scRT calculated from FACS-sorted mid S-phase mouse embryonic stem cells (mESC, left) and mouse neuroectoderm cells at day 7 of differentiation (mNE-7d, right). Bottom panels: binary scRT profiles of single cells sorted from top to bottom in function of the increasing replication percentage. Replicated and non-replicated regions are shown in red and blue, respectively. Upper panels: the pseudo-bulk RT calculated from the scRT profiles (see Methods) are compared with the bulk RT of the corresponding cell type. **b** Comparison between single-cell and bulk RT data. 2D density plot reporting pairwise comparisons between samples (colour code on the right) and Spearman correlations (circles) between RT data. Diagonal panels: RT distribution of each sample. **c** $T_{width}$ values of mESCs (upper panels) and mNE-7d (lower panels) cells for Early (RT > 0.5) and Late (RT ≤ 0.5) replicating regions. Higher $T_{width}$ indicates higher RT variability among cells. *P* values were calculated with the Kronos scRT Compare TW module (see "Methods"); **** < $10^{-4}$.

the whole cell population, 3.68 for sub-population 1, and 3.84 for sub-population 2 (Supplementary Fig. 3b). Due to the analysis resolution (~200 kb), we could only identify large-scale CNV differences between these sub-populations. Many of these alterations were on chromosomes 3, 7, 8, 11, 18 and 19 (Fig. 4b and Supplementary Fig. 3c). Particularly, in chromosome 3, the CN profiles suggested a clear chromosome-wide CN gain (Fig. 4b and Supplementary Fig. 3c) in sub-population 2 compared with sub-population 1. To further confirm this CN gain, we performed Fluorescence in situ Hybridization (FISH) on metaphase spreads with two DNA probes; one specific for the whole chromosome 3 and one for the centromere (Fig. 4c, d). Among the 344 spreads

analyzed, 162 (47.09%) had four copies and 170 (49.42%) had five copies of chromosome 3 (Fig. 4c–e), in agreement with our scCNV data (Fig. 4b and Supplementary Fig. 3c). Therefore, we could associate and normalize each S-phase group to its corresponding G1/G2-phase group and to individually compute scRT data for each MCF7 sub-population. This analysis demonstrates that, unlike population-based approaches, Kronos scRT can analyze the RT programme in heterogeneous cell populations and identify the existing sub-populations.

**scRT analysis in different human cell types**. Subsequently, we analyzed MCF7 cells as two individual sub-populations. Although

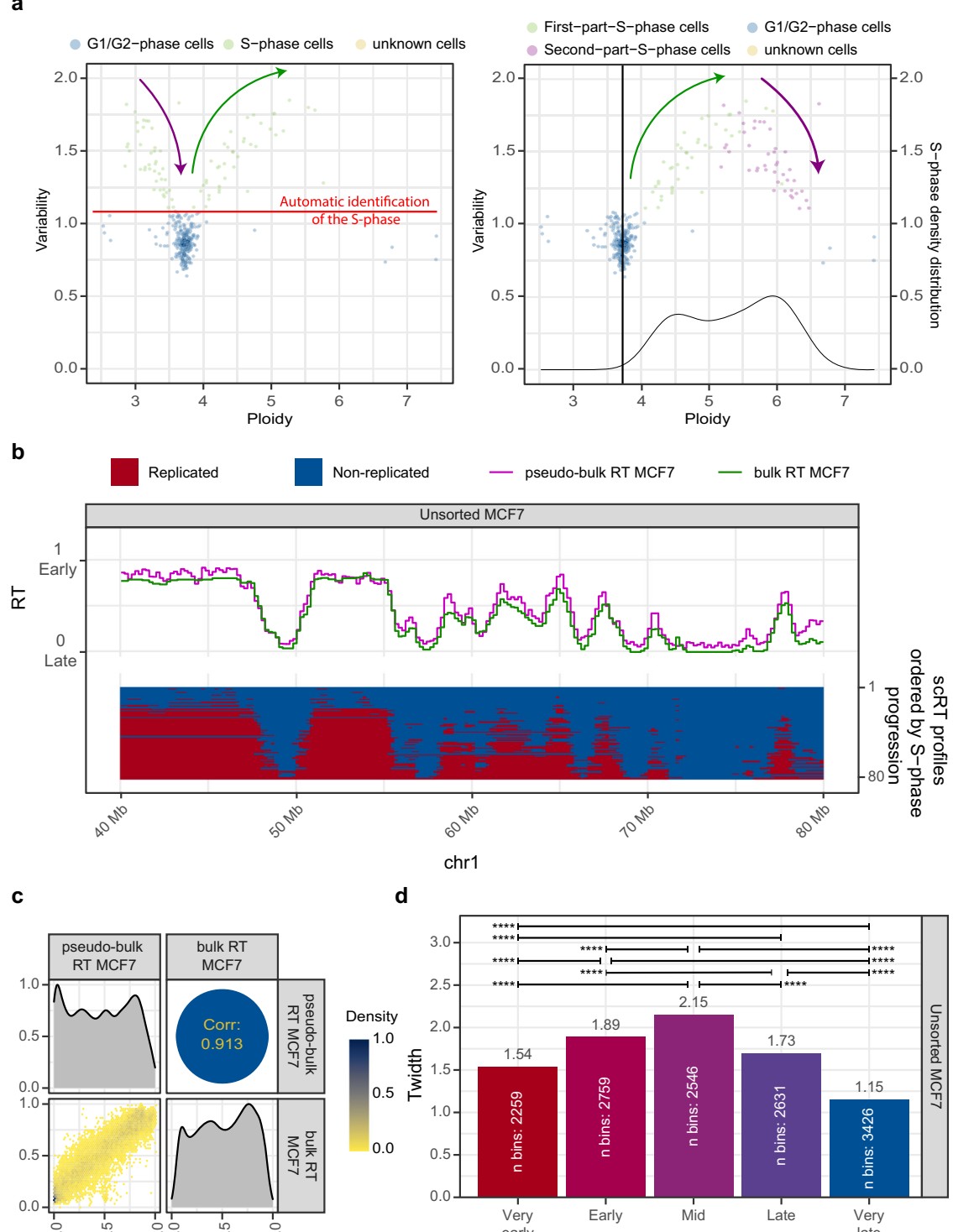

**Fig. 3 Extracting scRT from asynchronous human MCF7 cells. a** Scatter plots showing the intracellular bin-to-bin variability in function of single-cell ploidy before (left) and after (right) S-phase progression correction. The colour code is the same as in Fig. 1b, c. Kronos scRT can automatically identify the S-phase cells from the asynchronous scWGS data (see Methods). **b** scRT calculated from unsorted cycling MCF7 breast cancer cells. Same presentation as in Fig. 2a. **c** Comparison between single-cell and bulk RT data, as in Fig. 2b. **d** $T_{width}$ values calculated for genomic regions classified in five RT categories based on the pseudo-bulk RT values (Very early >0.8, 0.8≥ Early >0.6, 0.6≥ Mid >0.4, 0.4≥ Late >0.2, and Very Late ≤0.2). *P* values were calculated using the Kronos scRT Compare TW module ("Methods"); * < 0.05, ** < $10^{-2}$, *** < $10^{-3}$, **** < $10^{-4}$.

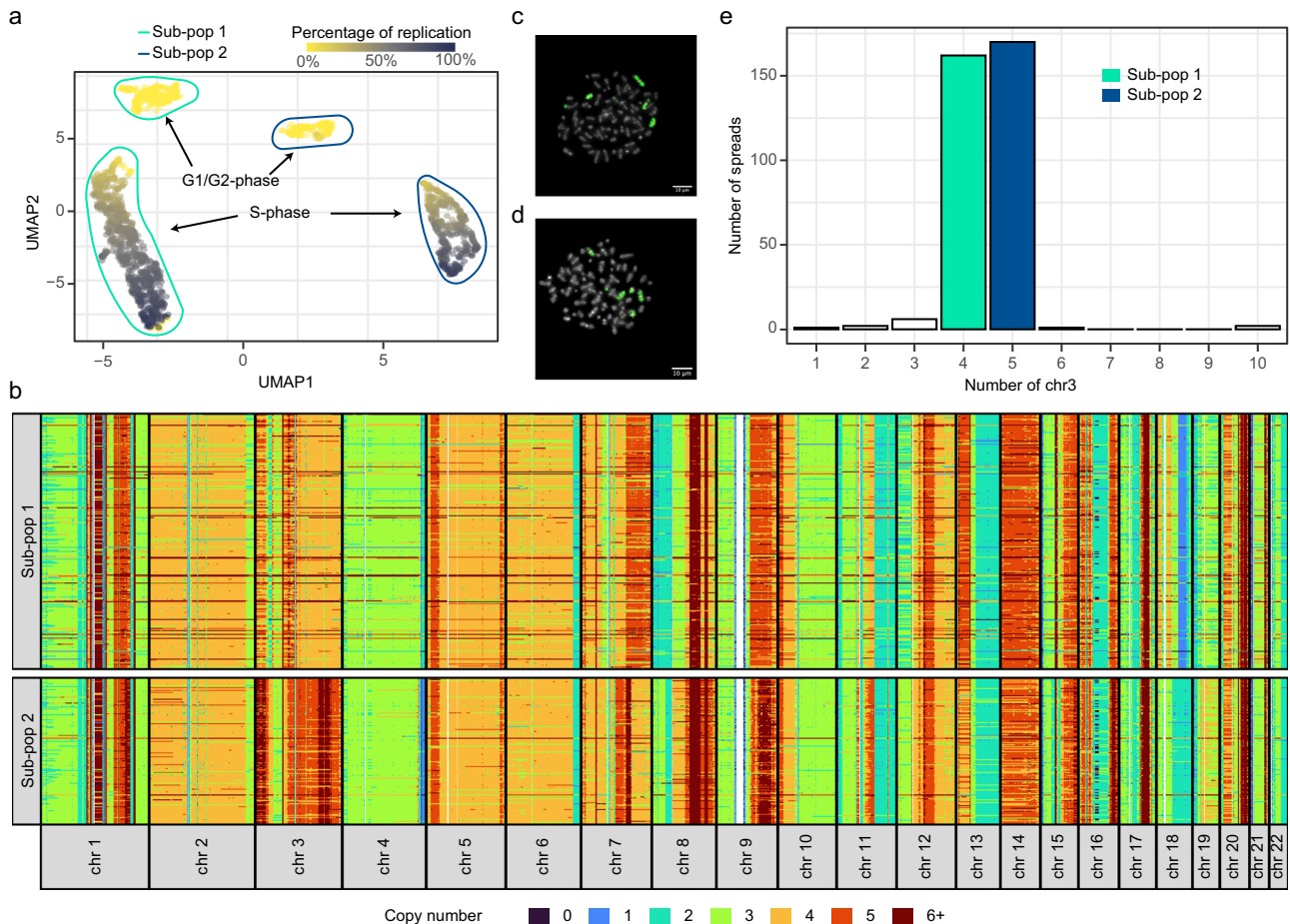

**Fig. 4 Identification of two sub-populations in MCF7 cell cultures. a** UMAP analysis of the scCNV data of MCF7 cells coloured in function of S-phase progression (only autosomal chromosomes). Both G1/G2 (yellow) and S-phase (colour intensity in function of genome replication as indicated) cells are separated into two major sub-populations. G1/G2- and S-phase cells are gated with colour-coded gates: aqua for sub-population 1 and blue for sub-population 2. **b** Copy number analysis of autosomes in G1/G2 MCF7 cells separated into two sub-populations based on the clustering results shown in (**a**). The binning for visualization is 1 Mb (median CN profile in Supplementary Fig. 3c). **c, d** Examples of FISH experiments for cells with 4 (**c**) and 5 (**d**) copies of chromosome 3 that correspond to sub-population 1 and sub-population 2, respectively. Chromosomes are stained with DAPI (grey), chromosome 3 is labelled in green, and its centromere in red. Scale bar: 10 μm. Two examples out of 344 cells obtained over two independent experiments. **e** Chromosome 3 count based on the FISH results (n=344). The two major groups of cells with 4 ($n = 162$) and 5 ($n = 170$) copies of chromosome 3 correspond to the two groups shown in (**a**): sub-population 1 and sub-population 2, respectively.

the pseudo-bulk RT of the two MCF7 sub-populations were very similar ($R = 0.946$) (Fig. 5a, b), we could distinguish the two sub-populations by UMAP analysis (Fig. 5c, d). We then generated and analyzed data for 514 HeLa cells (including 259 S-phase cells) and 1106 Jeff cells (normal lymphoblastoid cells, including 960 S-phase cells) (Supplementary Table 1). For each of these two cell lines, we identified only one population with median ploidy values of 2.87 (HeLa cells) and 1.94 (Jeff cells), respectively (Supplementary Fig. 3b). We calculated the scRT and pseudo-bulk RT profiles for HeLa and Jeff cells, as done for MCF7 cells (Supplementary Fig. 4a, d). The pseudo-bulk RT was highly correlated with the corresponding bulk RT in both cell types (Supplementary Figs. 2b and 4b, e). The scRT profiles were unique for each cell type and could be separated by dimensionality reduction analysis (Fig. 5c, d). Finally, we calculated the $T_{width}$ using the same five RT categories as done in Fig. 3d. For all analyzed cell types, the regions replicated at the very beginning or at very end of the S-phase were more synchronized (i.e. lower $T_{width}$ values ranging between 1.2 and 1.4 h in early and between 1.0 and 1.2 h in late-replicating regions) compared with regions replicated around mid S phase ($T_{width}$ values of 1.7-1.8 h) (Fig. 5e and Supplementary Fig. 4c, f).

**scRT of human cells shows stochastic variation within a cell population.** One of the main questions in the RT field is whether this programme is stochastic or deterministic. The authors of previous scRT studies commented on the improbability of the system to be stochastic because cell-to-cell variability was low[22,28]. However, their observations were based on limited cell numbers that reduce the possibility of identifying rare events.

We took advantage of the high number of scRT data obtained in the present study to tackle this issue. To better visualize stochasticity, we selected S-phase cells at three representative stages: early-S-phase cells (≤30% of replicated genome), mid-S-phase cells (40-60% of replicated genome) and late-S-phase cells (≥70% of replicated genome). We then assigned each 200 kb genomic bin to an RT category based on its pseudo-bulk RT, and for each bin, we calculated its replication probability at each representative stage (Fig. 6a, b and Supplementary Fig. 5a, b). If the RT programme were deterministic, we would not expect to see late-replicating regions (RT < 0.5) being replicated in early S-phase cells, and vice versa. As expected, in mid-S-phase cells, the majority of early-replicating regions were already replicated, while most late-replicating regions were not replicated yet (Fig. 6a, b and Supplementary Fig. 5a, b, middle panels). However, in all

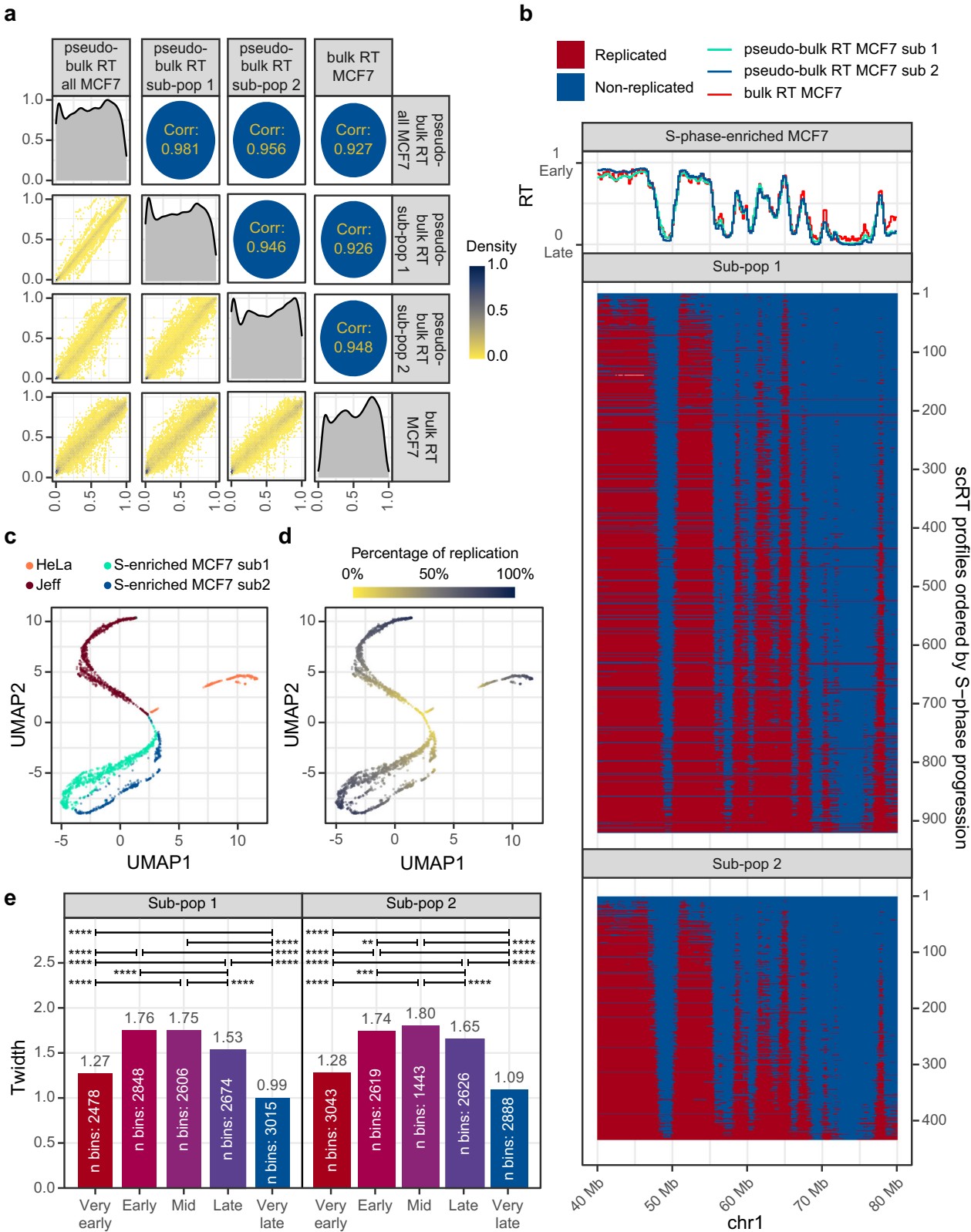

**Fig. 5 scRT analysis in S-phase enriched human cells. a** Pairwise comparison of the MCF7 cell pseudo-bulk RT and bulk RT profiles. Same as in Fig. 2b. **b** The scRT profiles of S-phase enriched MCF7 cells along a representative region. Upper: pseudo-bulk RT profiles of the two MCF7 sub-populations and bulk RT profile. Bottom: scRT profiles ordered from top to bottom in function of the genome replication percentage of each cell. **c, d** Dimensionality reduction analysis of scRT data for the indicated human cell lines (generated in the current study). Cells are colour-coded based on the cell type in (**c**) and based on the genome replication percentage in (**d**). **e** $T_{widths}$ calculated for the indicated five RT categories based on the pseudo-bulk RT values in the two MCF7 sub-populations. Categories were selected as in Fig. 3d. *P*-values were calculated using the Kronos scRT Compare TW module (see "Methods"); * < 0.05, ** < $10^{-2}$, *** < $10^{-3}$, **** < $10^{-4}$.

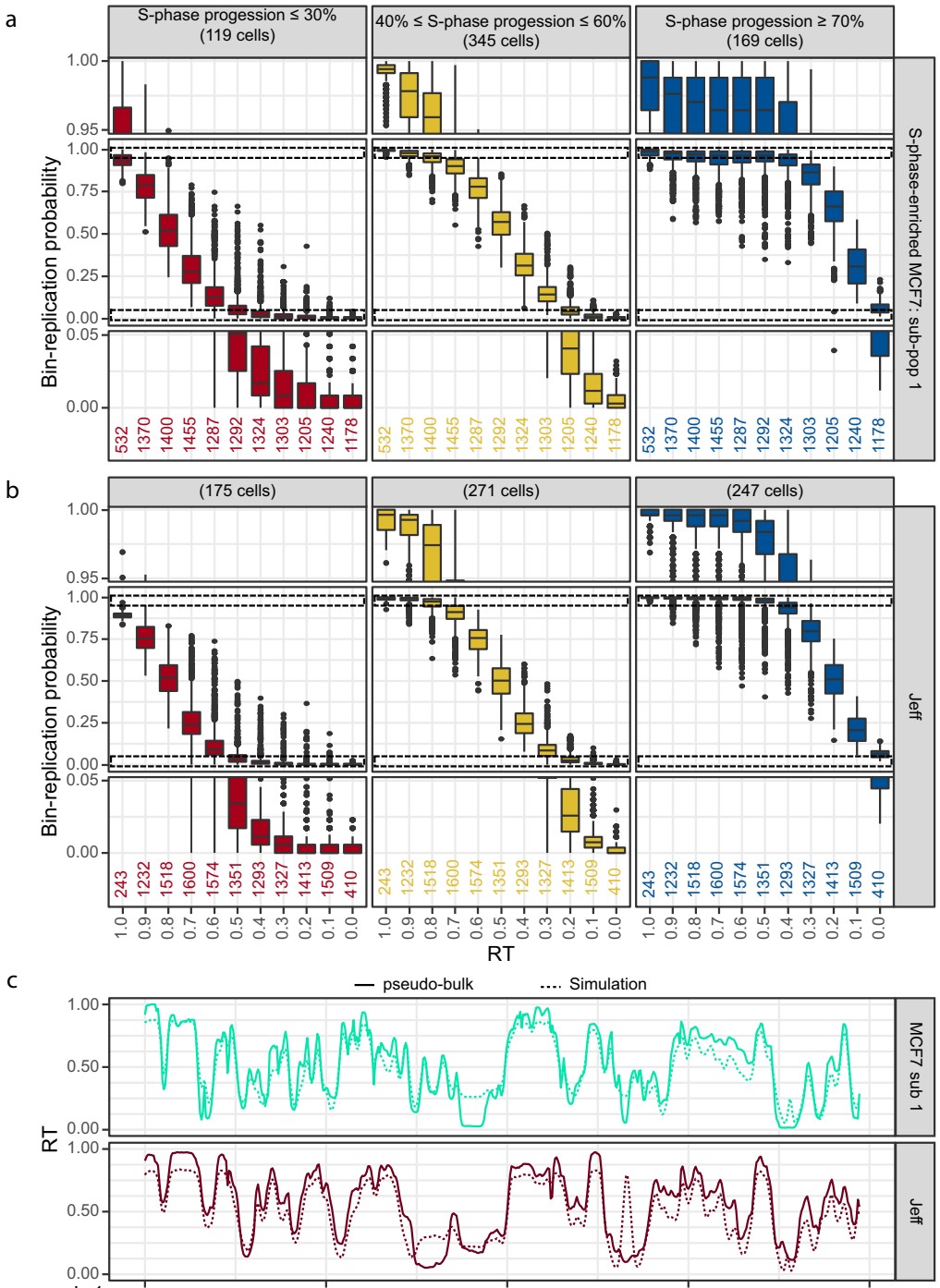

**Fig. 6 scRT data support a stochastic model of replication. a, b** Boxplots showing the replication probability (*y*-axis) relative to the pseudo-bulk RT (*x*-axis, 1 is early, and 0 is late) for S-phase-enriched MCF7 cells (sub-population 1) (**a**) and Jeff cells (**b**) at the indicated S-phase stages calculated in 200 kb bins. Middle panels: whole distribution; top and bottom panels: zoom on the areas showing the distribution upper and lower extremities, respectively (indicated with the dashed boxes in the middle panels). The numbers of genomic bins (n) are reported in the bottom panel below each boxplot. **c** Comparison between the pseudo-bulk RT (solid line) and simulated RT (dashed line) for the MCF7 cell sub-population 1 and Jeff cells. The simulation is based on Replicon[29] and uses the probability of being replicated in early S-phase cells (that completed up to 30% of their genome replication) for each 200 kb bin as input. Similar results were obtained with the scRT data of the S-phase-enriched MCF7 cell sub-population 2 and HeLa cells (Supplementary Fig. 5a, b, g). Spearman correlations are shown in Supplementary Fig. 5h.

the examined cell lines, we observed 1–5% of cells in which late genomic domains (pseudo-bulk RT < 0.5) were replicated at the beginning of the S phase (i.e. cells with ≤30% of replicated genome) (Fig. 6a, b and Supplementary Fig. 5a, b, left panels).

This was significantly higher than the value obtained for G1/G2 cells ($p < 10^{-6}$, one-sided paired Wilcoxon test), demonstrating that this is real biological variation rather than experimental noise (Supplementary Fig. 5c). According to the determinist model, in

late S-phase cells (cells with ≥70% of replicated genome), all early-replicating regions (pseudo-bulk RT > 0.5) should be completely replicated. However, this was not the case because only 95–99% of early bins were replicated at this stage (Fig. 6a, b and Supplementary Fig. 5a, b, right panels). We observed genomic regions replicated largely ahead or behind of schedule compared with the population average (i.e. late-replicating regions that are replicated in early-replicating cells, and vice versa) in most cells (Supplementary Fig. 5d). Moreover, these regions were not clustered into large domains (Supplementary Fig. 5e). This suggests that the observed out-of-schedule replications are not due to large CN gains or losses within individual cells. The probability that an out-of-schedule event was observed in two or more cells depended on the RT of the region, and extremely early- and late-replicating regions were more likely to exhibit unique events (Supplementary Fig. 5f).

It is important to note that in all examined cell types, the probability of a bin to be replicated in early S-phase cells was highly correlated with its population RT (Fig. 6a, b and Supplementary Fig. 5a, b, left panels). This suggests that the RT of a given genomic region depends on the origin firing probability in early S phase, as suggested by the stochastic models. To further test this hypothesis, we selected single cells at the beginning of the S phase (percentage of replicated genome ≤30%) and used the obtained scRT data to calculate the replication probability along the genome. We then used these results as an input for Replicon, a stochastic replication simulator[29,30], to simulate the RT programme throughout the genome (Fig. 6c and Supplementary Fig. 5g). The obtained simulated RT profiles were very similar (Spearman correlation ≥0.857) to the pseudo-bulk RT profiles of the corresponding cell lines or cell sub-populations (Fig. 6c and Supplementary Fig. 5g, h), demonstrating that the replication signals detected in early S-phase cells for late-replicating regions are real biological signals rather than technical noise. Our results strongly support the notion of a stochastic RT programme.

To evaluate the robustness of these results, we analyzed again the MCF7 sub-population 1 using three higher thresholds of the minimum number of reads required to keep a cell in the analysis. Increasing this limit allowed excluding cells with poorer CNV calling that could explain the observed variability. Regardless of the threshold used, the obtained results did not change, further supporting the robustness of the analysis (Supplementary Fig. 6a–c).

**Kronos scRT can extract scRT profiles from various single-cell DNA sequencing data.** Although we performed experiments using scWGS data that were generated for this scRT analysis, this sequencing technique is not a requirement for using Kronos scRT. As long as the single-cell sequencing data include CN information and concern cycling cells, Kronos scRT can process them and extract scRT profiles. Among the published datasets, the scHi-C data generated by Nagano et al.[31] are a perfect example to demonstrate this. The dataset concerns cycling mESCs grown on a feeder layer in medium with foetal bovine serum (mESC Serum) or without feeder but with PD and CHIR, two inhibitors that favour the maintenance of the mESC naive state (mESC 2i)[31]. Although these were paired-end sequencing data, due to their nature (i.e. read pairs can map to different genomic bins), we loaded them into Kronos scRT as single-end data. By doing so, we identified 312 G1/G2-phase and 329 S-phase cells in the mESC 2i sample, and 76 G1/G2-phase and 130 S-phase cells in the mESC Serum sample. We then calculated the scRT of S-phase cells and the pseudo-bulk RT profiles (Fig. 7a). These were highly correlated with the bulk RT and the pseudo-bulk RT of our scWGS data for sorted mid S-phase mESCs (Spearman

correlation >0.847) (Fig. 7a, b and Supplementary Fig. 2a). Furthermore, having more cells that are well distributed across the S phase allowed us to calculate the $T_{width}$ values with the same number of replication categories used for human samples (Fig. 7c). Our result confirmed that in both mESC samples, RT was tighter at the beginning and end of the S phase compared with the mid S phase, as observed in human cells. This suggests that it is an important common feature of mammalian cells.

## Discussion

RT studies have entered the single-cell era, but their conclusions have been limited by the small number of cells that can be analyzed. Here, we have overcome this scalability issue by using a microfluidic-based scWGS system to generate new data that we analyzed using our unified computational workflow Kronos scRT (Supplementary Fig. 1a).

We demonstrated that Kronos scRT allows the rapid extraction and analysis of scRT from various single-cell DNA sequencing datasets (e.g. scWGS, scHi-C) at the 200 kb resolution. It should be noted that the scRT analysis resolution depends on the scWGS data quality. For instance, scWGS data obtained with the multiple displacement amplification (MDA) approach, which has a strong GC bias[32], might require a much higher sequencing depth to provide reliable CNV calling, compared with other library preparation approaches with linear amplification, such as Linear Amplification via Transposon Insertion[20] and direct DNA transposition single-cell library preparation (DLP+)[33]. Importantly, Kronos scRT can directly extract scRT from sequencing data of asynchronously growing cells, without the time-consuming experimental procedures (e.g. cell sorting into G1 and S phase, manual processing on 96-well plates) required by other approaches[21–23]. Similarly, Massey and Koren[34] also used the 10x Genomics system to study scRT. Our down-sampling analysis indicates that with Kronos scRT, about 0.75 million reads are sufficient to obtain robust scRT data at a good resolution for a human diploid genome (Fig. 1d). On the basis of the current sequencing cost (~15 k€ for 10 G 100 bp paired-end reads on NovaSeq), the price is ~1€ per single cell. We successfully used Kronos scRT to obtain thousands of high-quality scRT profiles from various mouse and human cell types. This allowed us to study the stochastic replication events at an unprecedented depth. In agreement with recent ORM data using synchronized early S-phase cells[24,35], our results obtained directly from asynchronously growing cells also support a stochastic replication model. This indicates that the early replication events observed within the late-replicating regions detected by ORM are not due to activation of dormant origins upon cell synchronization.

Our analysis demonstrated that we can apply dimensionality reduction to scRT/scCNV profiles to identify cell sub-populations within heterogeneous samples. By normalizing the CN of S-phase MCF7 cell sub-populations relative to their corresponding G1/G2-phase sub-populations, we unveiled two different, although relatively similar, RT programmes (Fig. 5 and Supplementary Fig. 1d).

Moreover, dimensionality reduction analysis of scRT profiles gives a reconstruction of the replication timeline, from early to late S phase, in a given population, therefore forming pseudo-trajectories. Single-cell deconvolution of cell heterogeneity is an important factor to take into consideration, especially for data obtained from cancer samples, where normal and mutated cells coexist and mutated cells have undergone multiple rounds of random mutation and clonal expansion[33,36]. Furthermore, the possibility to identify sub-populations in a tissue might allow studying the RT programme of cells that cannot be cultured in vitro and for which specific markers are not available.

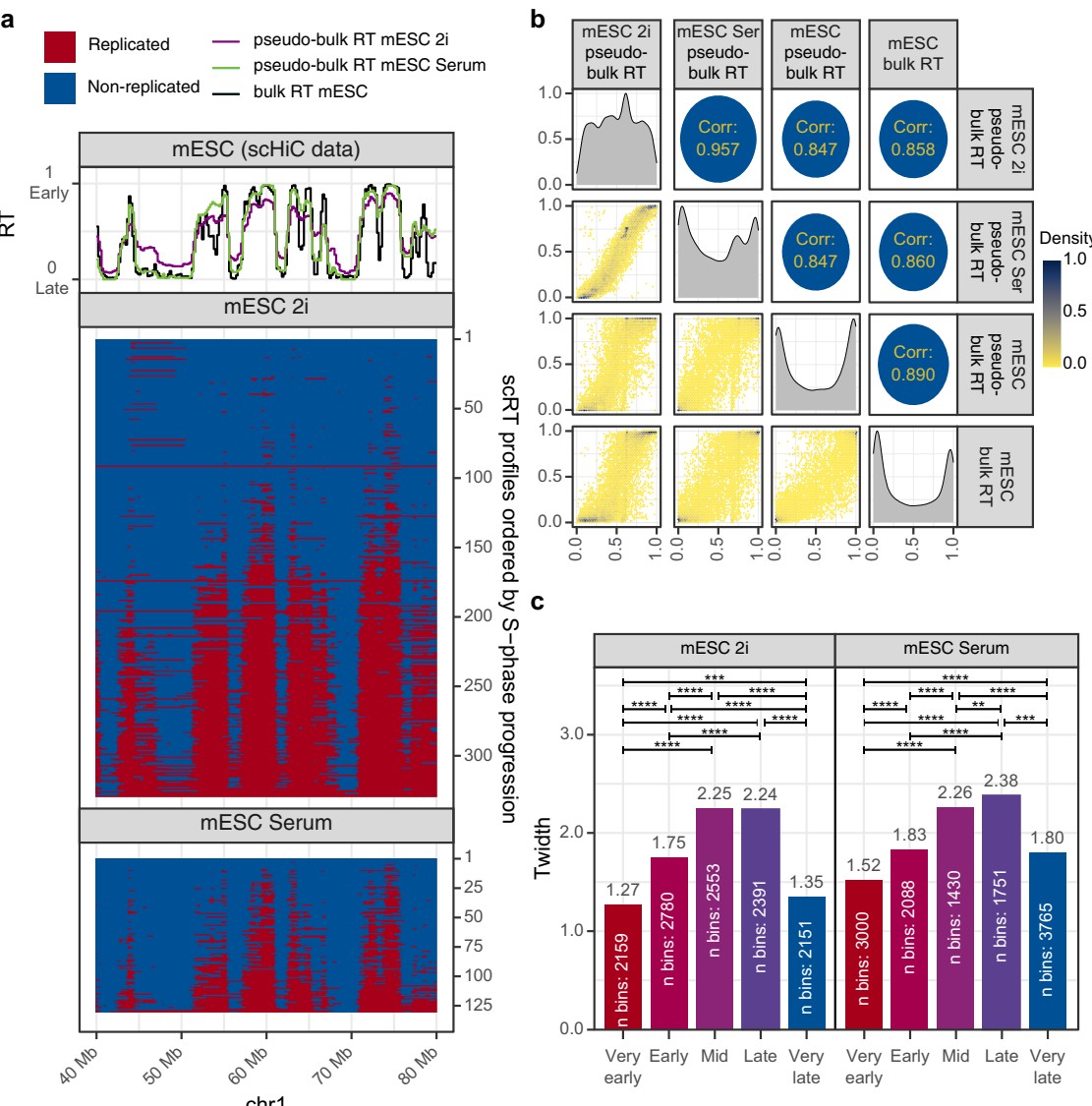

**Fig. 7 Kronos scRT analysis of scRT using scHi-C data. a** Top: pseudo-bulk RT profile of mESC grown in 2i (purple) or in serum (green) compared with bulk RT (black). Bottom: scRT profiles ordered, from top to bottom, in function of the genome replication percentage of each cell. **b** Pairwise comparisons of the pseudo-bulk RT and bulk RT profiles in mESC, mESC-2i, and mESC Serum. Same as in Fig. 2b. **c** $T_{width}$ calculated for the indicated five RT categories based on the corresponding pseudo-bulk RT values, as in Fig. 3d. $P$ values were calculated with the Compare TW module as before; * < 0.05, ** < $10^{-2}$, *** < $10^{-3}$, **** < $10^{-4}$.

Surprisingly, although the two MCF7 sub-populations identified in this study displayed significant CN differences (Fig. 4 and Supplementary Fig. 3), their RT profiles were highly correlated ($R = 0.946$) (Fig. 5). This suggests that the RT programme is an extremely robust process and that different copies of the same chromosome follow the same (or similar) replication programme. This is in agreement with previous reports showing that in the same cell type, the intracellular RT variation between homologues (using *Mus musculus* 129/Sv x *M. m. castaneus* hybrid cells) is similar to the intercellular variation observed between different cells[21]. Due to the low SNP coverage and low sequencing depth per single cell in our samples, we could not calculate the scRT of different homologues. Without separating the reads mapped on each allele, it is impossible to distinguish whether a bin has been replicated asynchronously between alleles (in the case of diploid cells), or if the two alleles are replicated synchronously while the region under study is not completely replicated. The situation is even more complex with aneuploid cell lines. Therefore, at the

current technical stage, we can only analyze scRT within each bin as a binary system (i.e. replicated and non-replicated). New algorithms need to be developed to obtain the haplotype-resolved scRT in order to explore the variation of replication programme between homologous chromosomes in normal human diploid cells or in cancer cells with complex karyotypes. This will help to better investigate the link between structural variations and RT changes, and its role in human diseases (e.g. during cancer development).

Additional studies are necessary to determine the molecular mechanisms that contribute to the degree of RT stochasticity. The combination of scRT and other single-cell omics data will provide additional insights into DNA replication regulation. It should be noted that RT has long been correlated with chromatin organization. Specifically, the RT data obtained in population and single-cell studies show that early/late-replicating domains are associated with A/B compartments;[21,23,37–40] although, the direct causal relationship remains unclear. Interestingly, although

changes in RT and compartments during mESC differentiation are tightly linked, they can be separated in specific contexts[23]. Comparison of haplotype-resolved bulk RNA-seq and scRT data from mESCs indicated that allelic replication asynchrony is frequently, but not always, associated with allelic expression[22]. Multiple mechanisms might cause extrinsic (cell-to-cell) and intrinsic (homologue-to-homologue) replication variability. Moreover, a major open question is whether the observed association between DNA replication and gene transcription results from transcription or from the open chromatin state of active genes[9,28]. Therefore, it is important to perform simultaneous multi-omic studies in the same single cells to better understand replication kinetics. Due to the remarkable simplicity of our tool that allows integrating higher cell numbers, Kronos scRT could be easily combined with the single-cell analysis of the transcriptome, DNA architecture, CpG methylation and chromatin accessibility[41–44], among others, offering the opportunity to study RT regulation at both a multi-omic scale and at the single-cell level. These single-cell multi-omic data could also be used to extract the RT landscapes, in addition to their original purpose (e.g. DNA architecture for scHi-C, chromatin accessibility for scATAC, transcription for scRNA-seq, etc). We think that it is important to increase the number of such studies to better comprehend DNA replication control and its stochasticity at the single-cell level.

## Methods

**Cell culture**. Oestrogen receptor-positive breast cancer MCF7 cells were cultured and treated as described in[45]. Briefly, cells were maintained in complete medium (DMEM supplemented with 10% FBS, 50 U/mL penicillin, and 50 μg/mL streptomycin) in 5% $CO_2$ at 37 °C. Oestrogen (E2) treatment was performed after hormone starvation. Cells were plated at ~25% confluency in complete medium for at least 16 h, rinsed thrice with PBS, and then hormone-starved for 48 h in DMEM without phenol red supplemented with 10% charcoal-stripped FBS (Dutscher), 50 U/mL penicillin, and 50 μg/mL streptomycin, before incubation with 100 nM E2 (dissolved in EtOH) for 24 h. Then, cells were harvested (70-80% confluence) by trypsin detachment.

HeLa S3 cells were cultured in DMEM high glucose medium with 10% FBS, and JEFF cells were grown in RPMI 1640 supplemented with 5% FBS. HeLa S3 cells were harvested by trypsin detachment at ~70–80% confluent, and JEFF cells were harvested at the density of $0.7–0.8 \times 10^6$ cells/mL by 200 g centrifugation for 5 min at room temperature.

**Sample preparation for single-cell copy number variation analysis**. Libraries were generated starting from exponentially growing cells. If mentioned, a step of S-phase enrichment by FACS sorting was performed. Cells were processed using the 10x Genomics Chromium single-cell CNV solution according to the manufacturer's instructions. Libraries were sequenced on an Illumina NovaSeq 6000 system using PE100, with the objective of obtaining ~2 million unique reads per single cell.

**S-phase enrichment by fluorescence-activated cell sorting**. Exponentially growing cells were collected and stained with 20 μg/mL Hoechst 33342 in complete medium at 37 °C for 1 h. Stained cells were rinsed twice with PBS, clumps were removed by passing each sample twice through a 40 μm strainer. The resulting single-cell suspension was stained with 50 μg/mL propidium iodide (PI) before FACS sorting. PI-positive cells were discarded. Cell-cycle stages were estimated in function of the Hoechst signal. Gates were positioned to collect S-phase cells and partially G1- and G2/M-phase cells. Sorted cells were collected in a 15 mL tube with 1 mL complete culture medium, rinsed once in PBS ($Ca^{2+}$ free, $Mg^{2+}$ free)/ 0.04% BSA and then processed with the 10x Genomics Chromium single-cell CNV solution kit, as previously described.

**Fluorescence in situ hybridization (FISH)**. For FISH analysis, cells were treated with colcemid (100 ng/ml, Roche) for 3 h and mitotic cells were collected by mitotic shake-off after a short trypsin treatment and centrifuged at 188 g for 10 min. Cell pellets were resuspended in 75 mM KCl and incubated in a 37 °C water bath for 15 min. Carnoy fixative solution (methanol:acetic acid, 3:1) was prepared and 1:10 volume added to the cells, before centrifugation at 188 g for 15 min. Cells were then fixed at room temperature in the Carnoy solution for 30 min, centrifuged and washed once more with the fixative solution. A minimum volume of fixative solution was left to resuspend the pellet and cells were dropped onto clean glass slides. FISH staining was performed following the manufacturer's instructions

(MetaSystems) using chromosome painting and centromere enumeration probes to specifically identify chromosome 3 (Metasystems probes). The Metafer imaging platform (MetaSystems) was used for automated acquisition of chromosome spread images. Picture triplets were merged with Fiji (v2.1.0) and the resulting images were manually scrutinized for chromosome 3 enumeration. Representative images were acquired using a Deltavision Core system (Applied Precision).

**10x Genomics data processing**. Data produced with the 10x Genomics system (various human cell lines) were processed using Cell Ranger DNA (10x Genomics software, version 1.0.0). The 10x Genomics subset-bam (version 1.0, https://github.com/10XGenomics/subset-bam) was used for bam file subsetting to obtain single-cell bam files. Duplicated reads were removed using Picard MarkDuplicates (version 2.6.0, http://broadinstitute.github.io/picard) and the resulting files were used as input for the CNV module of Kronos scRT (Supplementary Fig. 1a).

**Trimming and aligning reads**. The fastqtoBAM module of Kronos scRT uses demultiplexed fastq files as input and removes standard adaptors from reads. Adaptor trimming is performed using Trim Galore (version 0.4.4, https://www.bioinformatics.babraham.ac.uk/projects/trim_galore/), a modified version of cutadapt and FastQC. After trimming, reads are aligned to the provided reference genome (in our study, the human genome version hg38 and the mouse genome version mm10) using the RBowtie2 package (version 1.4.0) and only the best mapping for each read was reported. SAM files are then sorted, converted into BAM files using Rsamtools (version 1.34.1), and deduplicated using Picard MarkDuplicates (version 2.6.0). The fastqtoBam module was used to process all the mouse data analyzed in the current study.

**Calculate bin mappability and GC content for copy number estimation**. The binning module of Kronos scRT was used to calculate mappability and GC content in each genomic bin. This information is later used by the CNV module to normalize read counts and select the bins that will be considered for analysis. By default, the bin size is 20 kb, but it can be adjusted by the user in function of the sample mean sequencing depth. Moreover, only autosomal chromosomes are used by default, but the user can decide to keep also one or both sex chromosomes. GC content is calculated as the frequency of C and G in the reference sequence belonging to a bin.

To calculate mappability, this module simulates 1X coverage reads from a reference genome, adds mutations with an error rate of 0.1% (that can be adjusted by the user to fit the error rate of their datasets[46]) and maps the reads back to the reference genome using Rbowtie2 with the same settings used in the fastqtoBAM module. Read parameters (e.g. read length, single-end or paired-end reads, and fragment size) can be estimated from the BAM files of the single-cell experiment or manually set by the user. Then, the mappability of bin n ($M_n$) is calculated as the number of remapped reads of this bin ($Rr_n$), divided by the number of reads that were originally generated at the same location ($Rs_n$) (formula 1):

$$M_n = \frac{Rr_n}{Rs_n} \tag{1}$$

**CNV calling and intracellular bin-to-bin variability**. The CNV module of Kronos scRT counts the number of high-quality reads (i.e. mapping quality score ≥ 30) over the bins generated by the binning module. For CNV calculation of paired-end reads, if reads of a pair are mapped in the same bin, they are counted only once, otherwise, they are counted independently. Cells with $<2 \times 10^5$ reads are discarded (the user can manually adjust this threshold). Regions with a mappability <0.8 or >1.5 are excluded from the analysis (the user can also provide a list of blacklisted genomic regions and/or change the mappability thresholds). Read counts are then adjusted based on the mappability with formula (2):

$$rm_n = \frac{r_n}{M_n} \tag{2}$$

where $r_n$ is the read count on bin n, $M_n$ is the mappability of bin n obtained from (1) and $rm_n$ is the adjusted read count based on its mappability.

Read counts are then corrected for the GC content bias (formula 3):

$$R_n = \frac{rm_n \cdot \widetilde{rm_{GC|n}}}{\widetilde{rm}} \tag{3}$$

where $R_n$ is the normalized read count of bin n, $rm_n$ is the adjusted read counts (2), $GC|n$ represents all the bins with the same GC content as bin n, and the tilde represents the median.

The CNV module first calculates the bin-to-bin variability. To do so, it bins the genome into 500 kb bins and calculates the total read count for each bin. Then, it calculates the DIMAPD as defined in the 10x Genomics CNV Solution (https://support.10xgenomics.com/single-cell-dna/software). Assuming that the majority of analyzed cells are in the G1/G2 phase, in which the bin-to-bin variation is minimal, DIMAPD values are fitted to a Gaussian distribution, and cells that have significantly higher DIMAPD (formula 4, 5, 6 and 7) are considered to be in S

phase:

$$D_x = \frac{R_{i,x} - R_{i+1,x}}{\bar{R}} \qquad i \in [1, N] \qquad (4)$$

where $D_x$ is a vector containing differences between neighbouring 500 kb bins for cell x, $R_x$ is a vector containing the number of reads in 500 kb bins for cell x, and the index i identifies a bin (ranges between 1 to the total number of bins −1);

$$C_x = \sqrt{\sum_{i=1}^{N} \frac{R_{i,x}}{Gs}} \qquad (5)$$

where $C_x$ is the square root of the coverage, N is the number of 500 kb bins, $R_{i,x}$ is the number of reads in bin i in cell x, and Gs is the genome size in Mb;

$$MAPDC_x = \left| D_x - \widetilde{D_x} \right| \cdot C_x \qquad (6)$$

where $D_x$ comes from (4), and $C_x$ from (5).

As the MAPDC value increases linearly with the square root of the cell coverage, it was normalized as follows (formula 7) to obtain the DIMAPD:

$$DIMAPD_x = 1 + MAPDC_x - a\left(C_x - \widetilde{C}\right) - b \qquad (7)$$

where $MAPDC_x$ is defined in formula (6), $C_x$ in formula (5), and C is a vector containing the $C_x$ values for all cells in the experiment; a and b are two coefficients estimated through a linear fitting of MAPDC in function of the cell coverage distance from the median coverage of the experiment.

CNs are called starting from 20 kb bin tracks that are smoothed and segmented using a circular binary segmentation algorithm from the R package DNAcopy (version 1.56.0). Then, CNs are estimated by minimization of the following target function, as suggested by the 10x Genomics CNV Solution (https://support.10xgenomics.com/single-cell-dna/software; formula 8):

$$\chi = \sqrt{\sum_{n=1}^{N} S_n \cdot \sin^2 \frac{\pi \cdot R_n}{X}} \qquad (8)$$

where $S_n$ represents the size of segment n, $R_n$ represents the read count of segment n, and X is a number between the 5th and the 95th percentile of the read counts of all segments in a cell. Each local minimum of this equation is a possible solution to calculate CN. Therefore, a filter on local minimum values that lead to unreasonable mean ploidy (formula 10) is applied, and CN is calculated (formula 9). The filters used in this study can be found in Supplementary Table 2 (Ploidy limits). For sorted G1 and sorted mid S-phase cells in the mouse datasets, the ploidy value closer to 2 was selected:

$$CN_n = \left[ \frac{R_n}{X_{min}} \right] \qquad (9)$$

where $X_{min}$ is the value of X for $\vec{\chi}$ which is minimized (formula 7), $R_n$ is the read count in segment n, and $CN_n$ is an integer that represents the CN of segment n. The mean ploidy of a cell can then be calculated as follows (10):

$$P = \frac{\sum_{n=1}^{N} S_n \cdot CN_n}{\sum_{n=1}^{N} S_n} \qquad (10)$$

where P is the mean ploidy, $S_n$ is the size of segment n, and $CN_n$ is the copy number of segment n. The difference between the absolute minimum values and its closest relative minimum is used to evaluate how good the CN calling is. For values <2, the CN is not considered reliable (ploidy confidence). Negative values of ploidy confidence are imposed, as suggested by 10X Genomics. Bins included in the provided blacklist[47] are removed, as done here with the bins for mESCs and mNE-7d cells.

**Single-cell replication profiling and scRT calculation.** As already mentioned, the DIMAPD parameter can be used to distinguish replicating cells (in S phase) from non-replicating cells (in G1/G2 phase). The automatic threshold is a reasonable choice if the cell population has not been sorted for S-phase cell enrichment. For S-phase enriched cells, the diagnostic module can be used to manually select more adequate thresholds. The thresholds used in this study are reported in Supplementary Table 2. If available, FACS metadata can be integrated through the WhoIsWho module of Kronos scRT.

As shown in Fig. 1b, the function that we use to identify CN (formula 8) introduces some constraints in the calculation of the mean ploidy. Firstly, it is not possible to distinguish G1 from G2 cells that co-occupy the same area (Fig. 1b, blue population). Secondly, the S-phase cells are split into two groups (Fig. 1b, green population): the first group progresses normally, while the second group approaches the G1/G2 population from the left side of the plot, as indicated by the two arrows. Therefore, the Kronos scRT diagnostic module calculates two parameters to correct S-phase cell populations. Preferentially, the program tries to reunite all S-phase cells in a monomodal distribution in which the ploidy variability is maximized. When this is not possible, parameters are chosen to create a bimodal

distribution with a minimized ploidy variability. The user can manually set these parameters.

The CN of each segment is well corrected based on these values. According to our down-sampling (Fig. 1d and Supplementary Fig. 1b, c), cells with low coverage were filtered out. Coverage thresholds for each dataset are reported in Supplementary Table 2.

The genome is then binned again to calculate the scRT. Based on the sequencing depth of our samples, bins of 200 kb were used in our study. The bin size can be adjusted by the user. A weighted median CN is then calculated, where the weights are the sizes of overlaps between each 200 kb bin and the previously calculated segments.

The G1/G2-phase cell population was used to calculate a median pseudo-bulk CN profile that was used to normalize each S-phase cell as follows (formula 11):

$$nCN_{n,x} = \log_2\left( \frac{CN200_{n,x}}{\widetilde{CNG_n}} \right) \qquad (11)$$

where $nCN_{n,x}$ is the normalized copy number of bin n in the S-phase cell x, $CN200_{n,x}$ is the copy number of bin n in the S-phase cell x before normalization, and $CNG_n$ is the CN of bin n in all G1/G2-phase cells.

Each S-phase cell profile is then binarized. To do so, Kronos scRT identifies a value of nCN for which the following target function is minimized (12):

$$\varepsilon_{th,x} = \left( nCN_{n,x} - \begin{cases} 1 \text{ if } nCN_{n,x} \geq th \\ 0 \text{ if } nCN_{n,x} < th \end{cases} \right)^2 \qquad th \in [0, 1] \qquad (12)$$

where $\varepsilon_{th,x}$ is the Euclidian distance using the threshold th for cell x, and $nCN_{n,x}$ is obtained from (formula 11). Once the threshold that minimizes $\varepsilon_x$ is identified, scRT profiles can be calculated as follows (formula 13):

$$scRT_{n,x} = \begin{cases} 1 \text{ if } nCN_{n,x} \geq thm_x \\ 0 \text{ if } nCN_{n,x} < thm_x \end{cases} \qquad (13)$$

where $scRT_{n,x}$ is a binary value representing whether the bin n in cell x has been replicated (1) or not (0), $nCN_{n,x}$ comes from (11) and $thm_x$ is the th for each $\varepsilon_x$ minimized in cell x.

Simple matching coefficient distances are then calculated for each pair of cells. The population is filtered to remove cells that diverge by at least 25% from 60% of the single-cell population. Cells are then sorted in function of their genome replication percentage and tracks are averaged within each bin of replication percentage. To ensure a symmetrical distribution, outlier cells (i.e. very early and/or very late) are filtered out. Replication tracks per percentage interval are then averaged together to create the pseudo-bulk RT that is compared with the bulk RT. In this study, the coordinates of bulk RT of mESCs and NE-7d cells (issued from BrdU-IP samples) were converted to mm10 with the R package liftOver (v1.10). The bulk RT data of human cells were converted from hg19 to hg38 using the ucsc-liftOver tool (v366).

**Studying variability and sample differences.** To study cell-to-cell variability, Kronos RT calculates $T_{width}$ as defined in[21]: the time needed for genomic regions to be replicated in 25% to 75% of cells in a S phase lasting 10 h. The module Compare TW of Kronos RT allows users to apply a null hypothesis test by bootstrapping with H0 ($T_{width\_group1} = T_{width\_group2}$) and with H1 ($T_{width\_group1} \neq T_{width\_group2}$). For this, it randomly assigns the bins belonging to two groups to either of them, keeping the total original number of bins in each group constant. Newly assigned bins are then used to calculate the absolute difference between $T_{width\_group1}$ and $T_{width\_group2}$ that is then compared with the real difference (formula 14):

$$p = \frac{1}{N} \cdot \sum_{i=1}^{N} \begin{cases} 1 \text{ if } |Tw1_i - Tw2_i| \geq |T_{width1} - T_{width2}| \\ 0 \text{ if } |Tw1_i - Tw2_i| < |T_{width1} - T_{width2}| \end{cases} \qquad (14)$$

where p is the p value, N is the number of iterations (by default $10^4$), $Tw1_i$ and $Tw2_i$ are the $T_{width}$ calculated for the two groups in the iteration i, while $T_{width1}$ and $T_{width2}$ are the values of the real groups.

**Down-sampling.** To test the CN calling stability in function of the sequencing depth, G1/G2- and S-phase cells were selected from each experimental setting: i.e. 10x Genomics system, scWGA and scHi-C. Down-sampling was performed using Picard DownsampleSam (version 2.6.0, http://broadinstitute.github.io/picard). For each down-sampling coverage, cells with higher RPMb were used. In our study, RPMb thresholds were set as the value at which at least 75% of cells have a ploidy estimation that does not differ more than 5% from the original value. If the user does not define a specific threshold, Kronos scRT applies a threshold of 160 RPMb per haploid genome by default, which is close to the highest threshold we found for the different datasets analyzed in the current study (Fig. 1d and Supplementary Fig. 1b, c).

**Dimensionality reduction.** Kronos DRed is the dimensionality reduction module. This module uses genome-wide scCNV or scRT data to transform the data and provide a low-dimensional representation that reflects the important features (e.g. cell type, cell population, etc.). For CNV data, original values are used,

while for RT data, simple matching coefficient distances calculated with the R package ade4 (v1.7) are used. T-distributed Stochastic Neighbour Embedding (t-SNE)[48] and UMAP[26,27] are performed with the R packages Rtsne (v0.15) and umap (v0.2.7), respectively. For t-SNE, perplexity corresponds to a 50th of the number of cells or a minimum value of 10, theta to 0.25, and a partial principal component analysis is performed to calculate t-SNE coordinates over 5000 iterations. For binarized scRT data, the single matching coefficient distance matrix is provided to the Rtsne or umap functions (options input_mat='dist' for umap and is_distance=True for Rtsne). For CN data, the scCNV calling results from the RT module are provided (options input_mat='data' for umap and is_distance=False for Rtsne).

**MCF7 sub-population separation.** Kronos RT (option–extract_G1_G2_cells) was used to generate complete S- and G1/G2-phase CNV in 200 kb bins. Bins containing missing values or those belonging to sex chromosomes were removed. Dimensionality reduction of the resulting data was performed with UMAP using the R package umap (v0.2.7, option random_state=20210813). 2D UMAP coordinates were projected with the cell replication percentage to differentiate between G1/G2 and S phase. Cells were labelled based on manually attributed cut-off coordinates from the UMAP projection. For each resulting group, genome-wide scCNV data were visualized (Fig. 4b and Supplementary Fig. 3d). This allowed the manual attribution of S-phase cell groups to their corresponding G1/G2 cell groups and thus, the correct normalization of these groups in the downstream scRT analysis.

**Replication timing simulation.** The Replicon simulation code[29] was used to simulate the RT profiles. The Replicon simulator uses the initiation probability landscape (IPLS), i.e. the relative probability of initiating at any point in the genome, as input. In our simulations, the probability for each 200 kb of being replicated in early S-phase cells (that completed up to 30% of their genome replication) was based on the scRT data of the corresponding cell type. The same setting of other parameters as in our previous publication[24] was used, following the suggestion of the original study[30].

**Reporting summary.** Further information on research design is available in the Nature Research Reporting Summary linked to this article.

## Data availability
The mouse scWGS data were obtained from [https://www.ncbi.nlm.nih.gov/geo/query/acc.cgi?acc=GSE108556]. The mouse scHi-C data were obtained from [https://www.ncbi.nlm.nih.gov/geo/query/acc.cgi?acc=GSE94489]. The bulk RT data were obtained from [https://www.ncbi.nlm.nih.gov/geo/query/acc.cgi?acc=GSM923442] for MCF7 cells, [https://www.ncbi.nlm.nih.gov/geo/query/acc.cgi?acc=GSM923449] for HeLa cells, [https://www.ncbi.nlm.nih.gov/geo/query/acc.cgi?acc=GSM923451] for GM12878 cells (B-lymphoblastoid cell line) and [https://www.ncbi.nlm.nih.gov/geo/query/acc.cgi?acc=GSE108556] for mESC and NE-7d cells. The mm10 blacklist was obtained from https://github.com/Boyle-Lab/Blacklist. The raw and processed data generated in the current study has been deposited in NCBI's Gene Expression Omnibus (GEO) under accession number [https://www.ncbi.nlm.nih.gov/geo/query/acc.cgi?acc=GSE186173].

## Code availability
Kronos scRT is available in GitHub (https://github.com/CL-CHEN-Lab/Kronos_scRT) and archived in Zenodo[49].

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

## Acknowledgements

Work in C.L.C's laboratory is supported by the YPI program of Institut Curie, the ATIP-Avenir program from Centre national de la recherche scientifique (CNRS) and Plan Cancer from INSERM [grant number ATIP/AVENIR: No 18CT014-00], the CNRS 80| Prime interdisciplinary program, the Agence Nationale pour la Recherche (ANR) [grant number ReDeFINe − 19-CE12-0016-02, TELOCHROM − 19-CE12-0020-02] and Institut National du Cancer (INCa) [grant number PLBIO19-076]. J.M.J. is supported by a PSL-Qlife fellowship (ANR-17-CONV-0005). High-throughput sequencing was performed by the ICGex NGS platform of the Institut Curie supported by the grants ANR-10-EQPX-03 (Equipex) and ANR-10-INBS-09-08 (France Génomique Consortium) from the Agence Nationale de la Recherche ("Investissements d'Avenir" program), by the ITMO-Cancer Aviesan (Plan Cancer III) and by the SiRIC-Curie programme (SiRIC Grant INCa-DGOS- 465 and INCa-DGOS-Inserm_12554). D.F. receives salary support from the CNRS and Institut Curie. M.D.'s salary is covered by Emergences Grant from the City of Paris and the Cell biology and Cancer department of Institut Curie. The authors would like to acknowledge Geneviève Almouzni for MCF7 cells, the Institut Curie cell sorting facility for assistance with cell sorting, Dominika Foretek, Marc Gabriel and Ugo Szachnowski for stimulating discussions, and Elisabetta Andermarcher for critical reading of the manuscript.

## Author contributions

C.L.C. conceived and planned the study. X.W. performed the single-cell experiments with the assistance of M.B. and L.G.B. M.B. and L.G.B. performed the sequencing under the supervision of S.B. M.S. performed experiments and prepared the samples for FISH. M.D. performed the FISH experiment under the supervision of D.F. S.G. developed the program and J.M.J. contributed to its development. S.G. and J.M.J. performed the bioinformatics analyses. D.S. performed the RT simulation. C.L.C. supervised the experiments and bioinformatics analyses. S.G., J.M.J. and C.L.C. wrote the manuscript, and all the authors reviewed it.

## Competing interests

The authors declare no competing interests.
