## [Peer Review File · Nature Communications]

REVIEWER COMMENTS

Reviewer #1 (Remarks to the Author):

The authors developed a new software platform, Kronos scRT, that is able to perform single-cell replication timing analysis. Then they demonstrated this software by analyzing scWGS datasets in the public domain, as well as scWGS data generated in house specifically for this study. They found that most genomic regions in single cells replicate within a close time range around the replication timing measured in ensemble population average. Surprisingly, they also identified a small portion (1-5%) that replicate stochastically throughout the S phase, not following the replication timing of ensemble population average. While the software is of interested to the relevant community, thus worthwhile to be published, several aspects in the manuscript need to be improved before publication.

(1) Genome-wide bias and noise generated during single-cell WGA will affect the quality of scWGS data, thus may affect the performance of Kronos scRT analysis. For example, MDA-based method generally leads to a much higher genome-wide bias, thus not as suitable for CNV detection compared to some other scWGA methods. It is better for the authors to discuss a bit on this front, so that future readers will not be misled. Generally speaking, not all scWGS datasets are equal for Kronos scRT analysis. Some may not work as well, or at least require a different input number of sequencing reads per Mb per haploid genome, to achieve the same accuracy level for replication timing analysis. It will be ideal if the authors can try their software on different scWGS datasets in the public domain based on different scWGA methods, especially the MDA dataset, which may be worrisome.

(2) There is one section in the manuscript describing the capability of Kronos scRT software to identify sub-populations of cells within a heterogenous population. I noticed that this analysis is based on scCNV. The authors may want to clarify that not all types of heterogeneity in the population can be identified by this software, but only heterogeneity in large-scale CNV, or aneuploidy, as the authors demonstrated.

(3) I assume there is some level of flexibility for users to tweak the Kronos scRT software, such as tuning certain parameters when carrying out scRT analysis. These cutoff parameters will be largely arbitrary. I wonder the robustness of the analysis result. That is to say, will tuning such parameters affect the analysis outcome? Specifically, will tuning such parameters lead to any difference in scientific conclusions, especially the conclusion the authors put forward that “1-5% of the cells replicate late genomic domains in early S phase, while 1-5% of the early bins haven’t been replicated yet in late S phase cells”.

(4) Still focusing on the important scientific claim highlighted in the abstract that “replication can also occur stochastically throughout S phase”, referring to the final section of the main text. I found its supporting evidence not very strong, and wonder if can be further strengthened. The simulation merely support the notion that if the experimental evidence is true (1-5% of cells replicate late domains in early S phase, while 1-5% cells haven’t replicate early domains in late S phase), then a stochastic RT program will be a good fit. It will be ideal if the authors can further explore a bit along this line. Are these 1-5% of cells behaving differently from the population ensemble average in terms of RT still normal in other aspects? How about analyze many G1 and G2 cells as a negative control to rule out the possibility of

technical artifact such as WGA noise that may lead to the 1-5% of outliers? If such 1-5% outliers are true, it will be very informative to summarize some feature and statistics of such replication domains that have the potential to be replicated out of the population ensemble average RT window in 1-5% of cells. Any special feature such as their genomic locations or others?

Reviewer #2 (Remarks to the Author):

This study investigates the potential of single-cell DNA sequencing for advanced replication timing (RT) analysis of mammalian cells. Recent studies (refs 21-23) demonstrated that single cell DNA sequencing of FACS-sorted S phase cells allows to distinguish replicated from unreplicated genomic DNA segments due to the doubled copy number of replicated sequences.

The current study reports the elaboration of a comprehensive informatic analysis pipeline that can distinguish G1/G2 from S-phase cells based on intracellular bin-to-bin variability. Therefore, not only sorted cells but also unsorted asynchronous cell populations can be used, which can greatly simplify experimental procedures. The pipeline further allows to estimate the mean ploidy of the cells and establish their RT profile assuming that replicated regions have a doubled copy number compared to unreplicated regions, using G1/G2 cells as references. The pipeline can be applied to published single cell data that were determined for other purposes than RT analysis (e.g. single-cell Hi-C data). Combined with the 10x Genomics microfluidic system, the pipeline allowed the authors to analyse up to 1,000 single S-phase cells per cell line, which is about 10-fold more than in previous studies. Interestingly, they were able, using a data dimensionality reduction approach, to identify, within heterogeneous cell cultures, subpopulations with distinct chromosomal rearrangements and to establish and compare their respective RT profiles. Finally, their data allowed them to measure RT variability at different stages of S phase progression. The results support a stochastic model for replication.

Major criticisms

1. Although the technical quality of the benchwork and the bioinformatics is high, the increase in knowledge with respect to previous DNA replication studies is at best marginal. Many previous studies in yeast and mammalian cells, notably references 21-25, have already provided substantial evidence that both cell population and single-cell RT data are consistent with RT profiles emerging from independent and stochastic activation of replication origins, and have described how the distribution of RT around the mean varies from cell to cell at different stages of S phase. To me the only biological novelty is the clear identification of subpopulations with subtly different karyotypes and RT programs within cultures of single cell lines, particularly cancer cell lines.
2. If origin activation is stochastic, and if a locus can replicate with some probability at any time in S phase, as the authors suggest, there should be many instances where the two alleles of a locus (in diploid cells) replicate at different times. Therefore the copy number of any sequence may be 2, 3 or 4. However their pipeline only considers two possible states for a locus: replicated or unreplicated (i.e. either 2 or 4 copies, but not 3). This is furthermore problematic with highly aneuploid cell lines used in this study such as MCF-7, which has a mean ploidy of 3.64 with up to 5 copies of some chromosomes, so

that the copy number of any locus on this chromosome may be 5,6,7,8,9 or 10 and not just 5 or 10. Given that " cells with a RT profile deviating from the main population are filtered out" and that "excess cells (i.e. very early or very late) are filtered out", I suspect that the analyses mask the true heterogeneity of the replication program as they force the copy number profile to match a binarized partition between all copies replicated or all copies unreplicated. In this context, the sentence p.22 : "the RT program is an extremely robust process and different copies of the same chromosome follow similar replication program" is an obvious overstatement, given that the authors deliberately assume for their analysis that different chromosomal copies necessarily replicate in perfect coordination and are unable to detect the opposite situation. This is a severe limitation of the study that is neither discussed nor justified. The authors mention in the Discussion that it is not possible to resolve haplotypes but this is a different question that should not prevent a more nuanced copy number determination than either 2N or 4N.

If detecting replication of only a fraction of chromosomal copies is not doable, due for example to insufficient sequence coverage, please state and discuss this limitation clearly and specify how this problem could be solved in future analyses.

If on the contrary this determination is doable, please perform this analysis. Evaluating the proportion of cells in which all copies of a locus are not simultaneously replicated would be an important novel information, that cell-population techniques cannot capture, and that would be crucial for understanding the true cell-to-cell and homolog-to-homolog variability in RT.

3. Figure 1f was obscure to me. The axes were not labelled and I could not understand what I was looking at. How were cells ranked on the matrix rows and columns ? Furthermore I could not see any difference by eye between the two panels.

Minor criticisms

4. Twidth values are computed, if I understood correctly, assuming a 10h S phase. However no experimental measurements of S phase lengths in the different cell lines are provided. Where does this uniform estimated S phase duration come from ? Copy number profiles reveal the relative order of replication of different sequences but do not contain absolute temporal information. Wouldn't it be more rigorous to report Twidth variations as fractions of S phase length rather than as absolute times ?

5. Figure 2B and other similar Figures, lower triangle. Please remove the dark diagonal line as it masks the density values in this region of the graph and hampers the visualisation of the true spreading of the data around the diagonal.

6. Several sections of the Methods are referenced "as described by 10x Genomic" but no URL is given. I was unable to retrieve relevant information by surfing on the 10x Genomics site. Please give more precise references.

7. There are too many uncorrected typos and English language needs editing at times.

Reviewer #3 (Remarks to the Author):

Gnan et al report the development of a new methodology for the analysis of single cell replication timing. The method (Kronos scRT) takes advantage of the data generated by single cell DNA copy number mapping technologies such the 10X platform. Their uniform computational framework is able to process single cell data from a variety of inputs and generate large numbers of replication profiles for single cells. With their method, the authors are able to determine at which stage in S-Phase (or G1/G2) each cell is, and test how origin use differs across cells in the population. Gnan et al reveal that there is a high degree of consistency of origin firing in both early and late S-Phase, but less so in mid-S phase. In addition, the authors are able to define sub-populations of cells with different ploidy and address the long-standing question of whether replication patterns are deterministic or stochastic – their data indicates the latter.

Overall this is a very nice addition of the growing suite of tools to analyze single-cell data. The data is of high quality and is well presented. In particular Kronos scRT appears to be well suited to the analysis of datasets with comparatively low read numbers per cell, it should find good utility in the analysis of large datasets. I have no major criticisms with the paper.

REVIEWER COMMENTS

Reviewer #1 (Remarks to the Author):

The authors developed a new software platform, Kronos scRT, that is able to perform single-cell replication timing analysis. Then they demonstrated this software by analyzing scWGS datasets in the public domain, as well as scWGS data generated in house specifically for this study. They found that most genomic regions in single cells replicate within a close time range around the replication timing measured in ensemble population average. Surprisingly, they also identified a small portion (1-5%) that replicate stochastically throughout the S phase, not following the replication timing of ensemble population average. While the software is of interested to the relevant community, thus worthwhile to be published, several aspects in the manuscript need to be improved before publication.

We would like to thank the reviewer for his/her positive support of our study.

(1) Genome-wide bias and noise generated during single-cell WGA will affect the quality of scWGS data, thus may affect the performance of Kronos scRT analysis. For example, MDA-based method generally leads to a much higher genome-wide bias, thus not as suitable for CNV detection compared to some other scWGA methods. It is better for the authors to discuss a bit on this front, so that future readers will not be misled. Generally speaking, not all scWGS datasets are equal for Kronos scRT analysis. Some may not work as well, or at least require a different input number of sequencing reads per Mb per haploid genome, to achieve the same accuracy level for replication timing analysis. It will be ideal if the authors can try their software on different scWGS datasets in the public domain based on different scWGA methods, especially the MDA dataset, which may be worrisome.

We agree with the reviewer that the performance of scRT analysis depends on the scWGS data quality. Unfortunately, we could not find an MDA dataset of asynchronously growing cells that contains enough cells with sufficient sequencing depth suitable for the scRT analysis.

To better discuss the possible effects of scWGS quality and sequencing depth on scRT data analysis, we modified the following text in the Results section:

“The minimum number of reads required for CNV calling depends on the experimental settings and the procedure used to create the scWGS libraries. Therefore, for each dataset, it is important to select G1/G2- and S-phase cells with relatively high coverage to determine by down-sampling the robustness of CNV detection and the minimum number of reads required for correct CNV calling.” Page 5, line 35 and Page 6 lines 1-3.

In addition, we added the following sentences in the Discussion section:

“It should be noted that the scRT analysis resolution depends on the scWGS data quality. For instance, scWGS data obtained with the multiple displacement amplification (MDA) approach, which has a strong GC bias³², might require a much

higher sequencing depth to provide reliable CNV calling, compared with other library preparation approaches with linear amplification, such as Linear Amplification via Transposon Insertion (LIANTI)²⁰ and direct DNA transposition single-cell library preparation (DLP+)³³.” Page 11, lines 22-38.

(2) There is one section in the manuscript describing the capability of Kronos scRT software to identify sub-populations of cells within a heterogenous population. I noticed that this analysis is based on scCNV. The authors may want to clarify that not all types of heterogeneity in the population can be identified by this software, but only heterogeneity in large-scale CNV, or aneuploidy, as the authors demonstrated.

Indeed, this analysis to identify sub-populations is initially based on scRT and not scCNV. This is now specified in the following sentences:

“While performing the dimensionality reduction analysis with Uniform Manifold Approximation and Projection (UMAP)^{26,27} on the scRT profiles of the S-phase enriched MCF7 cells, we noticed that cells were distributed into two distinct trajectories in which cells were sorted according to their S-phase progression (Supplementary Fig. 3a).” Page 8, lines 15-18.

After identifying these two sub-populations based on scRT and knowing that the karyotype of MCF7 cells can be quite unstable, we decided to further characterize our MCF7 cell data using scCNV. We determined that the two sub-populations show two different karyotypes.

To be more specific about the differences that can be identified in our study, we modified the main text as follows:

“Due to the analysis resolution (~200 kb), we could only identify large-scale CNV differences between these sub-populations. Many of these alterations were on chromosomes 3, 7, 8, 11, 18 and 19 (Fig. 4b and Supplementary Fig. 3c).” Page 8, lines 24-27.

(3) I assume there is some level of flexibility for users to tweak the Kronos scRT software, such as tuning certain parameters when carrying out scRT analysis. These cutoff parameters will be largely arbitrary. I wonder the robustness of the analysis result. That is to say, will tuning such parameters affect the analysis outcome? Specifically, will tuning such parameters lead to any difference in scientific conclusions, especially the conclusion the authors put forward that “1-5% of the cells replicate late genomic domains in early S phase, while 1-5% of the early bins haven’t been replicated yet in late S phase cells”.

We agree with the reviewer that modifying certain parameters, e.g. minimum coverage, ploidy limits, G1/G2-phase and S-phase thresholds, among others (see Supplementary Table 2 for a list of all parameters used in our study) might affect the results. These cutoff parameters are arbitrary to some extent. However, once they have been set within reasonable ranges, slightly tuning them will not cause major impact in the analysis outcome and the scientific conclusions. This is because only a few cells (<5%) will be affected. Amongst these parameters, the threshold for the minimal number of reads to include a cell in the analysis is a parameter that can affect most the analysis. Therefore, we decided to address the robustness of our result by

testing three higher cut-offs (130, 145, and 160 reads per Mb per haplotype) and compared them with the one used in the analyses for the S-phase enriched MCF7 cells sub-population 1. The key results (i.e. pseudo-bulk RT, T-width, and Bin-replication probability) were confirmed and the scientific conclusions of our study were not affected by these changes. The results of this analysis can be found in the new Supplementary Figure S6.

We modified the text to include the new results as follows: “To evaluate the robustness of these results, we analysed again the MCF7 sub-population 1 using three higher thresholds of the minimum number of reads required to keep a cell in the analysis. Increasing this limit allowed excluding cells with poorer CNV calling that could explain the observed variability. Regardless of the threshold used, the obtained results did not change, further supporting the robustness of the analysis (Supplementary Fig. 6a-c).” Page 10, lines 27-31.

(4) Still focusing on the important scientific claim highlighted in the abstract that “replication can also occur stochastically throughout S phase”, referring to the final section of the main text. I found its supporting evidence not very strong, and wonder if can be further strengthened. The simulation merely support the notion that if the experimental evidence is true (1-5% of cells replicate late domains in early S phase, while 1-5% cells haven't replicate early domains in late S phase), then a stochastic RT program will be a good fit. It will be ideal if the authors can further explore a bit along this line. Are these 1-5% of cells behaving differently from the population ensemble average in terms of RT still normal in other aspects? How about analyze many G1 and G2 cells as a negative control to rule out the possibility of technical artifact such as WGA noise that may lead to the 1-5% of outliers? If such 1-5% outliers are true, it will be very informative to summarize some feature and statistics of such replication domains that have the potential to be replicated out of the population ensemble average RT window in 1-5% of cells. Any special feature such as their genomic locations or others?

We thank the reviewer for the great suggestion of using G1/G2 cells as a negative control. We did the corresponding analysis and showed that the observed results are not due to technical artifacts. The results of this analysis can be found in the new Supplementary Figure 5c.

We modified the text to include the new results as follows: “However, in all the examined cell lines, we observed 1-5% of cells in which late genomic domains (pseudo-bulk RT <0.5) were replicated at the beginning of the S phase (i.e. cells with ≤30% of replicated genome) (Fig. 6a,b and Supplementary Fig. 5a,b, left panels). This was significantly higher than the value obtained for G1/G2 cells ($p < 10^{-6}$, one-sided paired Wilcoxon test), demonstrating that this is real biological variation rather than experimental noise (Supplementary Fig. 5c).” Page 9, lines 33-36 and Page 10, lines 1-2.

We then studied the features of these out-of-schedule bins. For each bin within the late-replicating domains, there are 1-5% of early S-phase cells in which this bin is replicated. However, all these out-of-schedule replicating bins do not belong to the same cells. Rather, they are scattered across early/late replicating cells. To address

this point, we added three panels to Supplementary Figure 5: d-f. In panel S5d, we looked at the number of unscheduled events per cell, and found that most cells have multiple unscheduled events. In panel S5e, we looked at the number of adjacent bins that we considered as out of schedule, and we showed that they are not clustering. In panel S5f, we counted how many cells contain a certain bin replicated out of schedule, and we plotted the distribution of these out-of-schedule bins according to the number of cells that share this event. These plots show that most events are unique or shared among few (2 or 3) cells.

To clarify this part, we modified the text as follows: “We observed genomic regions replicated largely ahead or behind of schedule compared with the population average (i.e. late-replicating regions that are replicated in early-replicating cells, and vice versa) in most cells (Supplementary Fig. 5d). Moreover, these regions were not clustered into large domains (Supplementary Fig. 5e). This suggests that the observed out-of-schedule replications are not due to large CN gains or losses within individual cells. The probability that an out-of-schedule event was observed in two or more cells depended on the RT of the region, and extremely early- and late-replicating regions were more likely to exhibit unique events (Supplementary Fig. 5f).” Page 10, lines 6-13.

Reviewer #2 (Remarks to the Author):

This study investigates the potential of single-cell DNA sequencing for advanced replication timing (RT) analysis of mammalian cells. Recent studies (refs 21-23) demonstrated that single cell DNA sequencing of FACS-sorted S phase cells allows to distinguish replicated from unreplicated genomic DNA segments due to the doubled copy number of replicated sequences. The current study reports the elaboration of a comprehensive informatic analysis pipeline that can distinguish G1/G2 from S-phase cells based on intracellular bin-to-bin variability. Therefore, not only sorted cells but also unsorted asynchronous cell populations can be used, which can greatly simplify experimental procedures. The pipeline further allows to estimate the mean ploidy of the cells and establish their RT profile assuming that replicated regions have a doubled copy number compared to unreplicated regions, using G1/G2 cells as references. The pipeline can be applied to published single cell data that were determined for other purposes than RT analysis (e.g. single-cell Hi-C data). Combined with the 10x Genomics microfluidic system, the pipeline allowed the authors to analyse up to 1,000 single S-phase cells per cell line, which is about 10-fold more than in previous studies. Interestingly, they were able, using a data dimensionality reduction approach, to identify, within heterogeneous cell cultures, subpopulations with distinct chromosomal rearrangements and to establish and compare their respective RT profiles. Finally, their data allowed them to measure RT variability at different stages of S phase progression. The results support a stochastic model for replication.

Major criticisms

1. Although the technical quality of the benchwork and the bioinformatics is high, the increase in knowledge with respect to previous DNA replication studies is at best

marginal. Many previous studies in yeast and mammalian cells, notably references 21-25, have already provided substantial evidence that both cell population and single-cell RT data are consistent with RT profiles emerging from independent and stochastic activation of replication origins, and have described how the distribution of RT around the mean varies from cell to cell at different stages of S phase. To me the only biological novelty is the clear identification of subpopulations with subtly different karyotypes and RT programs within cultures of single cell lines, particularly cancer cell lines.

We thank the reviewer for acknowledging the high quality of both our benchwork and bioinformatics analysis. The reviewer also pointed out one of the important novelties of our study: "clear identification of subpopulations with different karyotypes and RT programs issued from single cultures of single cell lines", which in our view is an essential foundation to further study the RT programs within heterogeneous cell populations, such as tumors.

2. If origin activation is stochastic, and if a locus can replicate with some probability at any time in S phase, as the authors suggest, there should be many instances where the two alleles of a locus (in diploid cells) replicate at different times. Therefore the copy number of any sequence may be 2, 3 or 4. However their pipeline only considers two possible states for a locus: replicated or unreplicated (i.e. either 2 or 4 copies, but not 3). This is furthermore problematic with highly aneuploid cell lines used in this study such as MCF-7, which has a mean ploidy of 3.64 with up to 5 copies of some chromosomes, so that the copy number of any locus on this chromosome may be 5,6,7,8,9 or 10 and not just 5 or 10. Given that "cells with a RT profile deviating from the main population are filtered out" and that "excess cells (i.e. very early or very late) are filtered out", I suspect that the analyses mask the true heterogeneity of the replication program as they force the copy number profile to match a binarized partition between all copies replicated or all copies unreplicated. In this context, the sentence p.22 : "the RT program is an extremely robust process and different copies of the same chromosome follow similar replication program" is an obvious overstatement, given that the authors deliberately assume for their analysis that different chromosomal copies necessarily replicate in perfect coordination and are unable to detect the opposite situation. This is a severe limitation of the study that is neither discussed nor justified. The authors mention in the Discussion that it is not possible to resolve haplotypes but this is a different question that should not prevent a more nuanced copy number determination than either 2N or 4N. If detecting replication of only a fraction of chromosomal copies is not doable, due for example to insufficient sequence coverage, please state and discuss this limitation clearly and specify how this problem could be solved in future analyses. If on the contrary this determination is doable, please perform this analysis. Evaluating the proportion of cells in which all copies of a locus are not simultaneously replicated would be an important novel information, that cell-population techniques cannot capture, and that would be crucial for understanding the true cell-to-cell and homolog-to-homolog variability in RT.

As suggested by the reviewer, we further discussed this limitation in the Discussion section and specified that this problem needs to be solved in future analyses.

"Due to the low SNP coverage and low sequencing depth per single cell in our samples, we could not calculate the scRT of different homologues. Without separating the reads mapped on each allele, it is impossible to distinguish whether a bin has been replicated

asynchronously between alleles (in the case of diploid cells), or if the two alleles are replicated synchronously while the region under study is not completely replicated. The situation is even more complex with aneuploid cell lines. Therefore, at the current technical stage, we can only analyse scRT within each bin as a binary system (i.e. replicated and non-replicated). New algorithms need to be developed to obtain the haplotype-resolved scRT in order to explore the variation of replication programme between homologous chromosomes in normal human diploid cells or in cancer cells with complex karyotypes.” Page 12, lines 26-35.

3. Figure 1f was obscure to me. The axes were not labelled and I could not understand what I was looking at. How were cells ranked on the matrix rows and columns ? Furthermore I could not see any difference by eye between the two panels.

In Figure 1f of our original manuscript, the plots show an example of simple matching coefficient matrix among single cells before (left panel) and after (right panel) filtering. The cells were ordered accordingly to their S-phase progression (i.e. percentage of genome replication). The irregular cells with a low matching coefficient with their neighbors were discarded. The filtering that we performed was very mild and only few cells were excluded. Therefore, we agree with the reviewer that this figure is not very informative and decided to remove the plot in our revised manuscript.

Minor criticisms

4. Twidth values are computed, if I understood correctly, assuming a 10h S phase. However no experimental measurements of S phase lengths in the different cell lines are provided. Where does this uniform estimated S phase duration come from ? Copy number profiles reveal the relative order of replication of different sequences but do not contain absolute temporal information. Wouldn't it be more rigorous to report Twidth variations as fractions of S phase length rather than as absolute times ?

We agree completely with the reviewer about the fact that scRT data based on scWGS data do not provide any real temporal information about the S-phase length in hours. The definition of Twidth and the usage of 10 h has been adopted from Dileep, Vishnu, and David M. Gilbert. 2018. “Single-Cell Replication Profiling to Measure Stochastic Variation in Mammalian Replication Timing.” *Nature Communications* 9(1): 427. <http://www.nature.com/articles/s41467-017-02800-w> (November 9, 2018), which converts a 10% genome replication in 1 h. This arbitrary conversion of the percentage of replicated genome into hours makes it easier for most of the readers to grasp the way of calculation and the meaning of the Twidth.

To clarify we modified the main text as follows “We then wanted to quantify the replication variability within each cell population. As scRT data do not provide precise information on the actual time needed for genome replication, we decided to quantify scRT variability using the concept of T_{width} introduced in a recent scRT study²¹. T_{width} is defined as the time needed for a given genomic region to be replicated from 25% to 75% of cells in an S phase lasting 10 h. Although S-phase length is not the same in all cells, assuming a uniform 10h S-phase length makes it easy to compare results of different datasets and results obtained in previous studies.” Page 7, lines 12-18.

5. Figure 2B and other similar Figures, lower triangle. Please remove the dark diagonal line as it masks the density values in this region of the graph and hampers the visualisation of the true spreading of the data around the diagonal.

We have made the modifications suggested by the reviewer.

6. Several sections of the Methods are referenced "as described by 10x Genomic" but no URL is given. I was unable to retrieve relevant information by surfing on the 10x Genomics site. Please give more precise references.

As suggested by the reviewer, we added the corresponding URL (<https://support.10xgenomics.com/single-cell-dna/software>) at page 16, line 17, and page 17, line 7.

7. There are too many uncorrected typos and English language needs editing at times.

The revised manuscript has been proof-read by native English speakers.

Reviewer #3 (Remarks to the Author):

Gnan et al report the development of a new methodology for the analysis of single cell replication timing. The method (Kronos scRT) takes advantage of the data generated by single cell DNA copy number mapping technologies such the 10X platform. Their uniform computational framework is able to process single cell data from a variety of inputs and generate large numbers of replication profiles for single cells. With their method, the authors are able to determine at which stage in S-Phase (or G1/G2) each cell is, and test how origin use differs across cells in the population. Gnan et al reveal that there is a high degree of consistency of origin firing in both early and late S-Phase, but less so in mid-S phase. In addition, the authors are able to define sub-populations of cells with different ploidy and address the long-standing question of whether replication patterns are deterministic or stochastic – their data indicates the latter.

Overall this is a very nice addition of the growing suite of tools to analyze single-cell data. The data is of high quality and is well presented. In particular Kronos scRT appears to be well suited to the analysis of datasets with comparatively low read numbers per cell, it should find good utility in the analysis of large datasets. I have no major criticisms with the paper.

We would like to thank the reviewer for his/her positive evaluation and support.

REVIEWERS' COMMENTS

Reviewer #1 (Remarks to the Author):

The authors have addressed all of my concerns and suggestions. I have no further issues to raise with this paper.

Reviewer #2 (Remarks to the Author):

I am satisfied with the authors' answers to my comments.